# Sexual dimorphism in striatal dopaminergic responses promotes monogamy in social songbirds

**Kirill Tokarev**[1,2]*, **Julia Hyland Bruno**[1,3], **Iva Ljubičić**[1,4], **Paresh J Kothari**[2], **Santosh A Helekar**[5], **Ofer Tchernichovski**[1,3,4], **Henning U Voss**[2]

[1]Department of Psychology, Hunter College, City University of New York, New York, United States; [2]Department of Radiology, Weill Cornell Medicine, New York, United States; [3]Department of Psychology, Graduate Center of the City University of New York, New York, United States; [4]Department of Biology, Graduate Center of the City University of New York, New York, United States; [5]Department of Neurology, Houston Methodist Research Institute, Houston, United States

**Abstract** In many songbird species, males sing to attract females and repel rivals. How can gregarious, non-territorial songbirds such as zebra finches, where females have access to numerous males, sustain monogamy? We found that the dopaminergic reward circuitry of zebra finches can simultaneously promote social cohesion and breeding boundaries. Surprisingly, in unmated males but not in females, striatal dopamine neurotransmission was elevated after hearing songs. Behaviorally too, unmated males but not females persistently exchanged mild punishments in return for songs. Song reinforcement diminished when dopamine receptors were blocked. In females, we observed song reinforcement exclusively to the mate's song, although their striatal dopamine neurotransmission was only slightly elevated. These findings suggest that song-triggered dopaminergic activation serves a dual function in social songbirds: as low-threshold social reinforcement in males and as ultra-selective sexual reinforcement in females. Co-evolution of sexually dimorphic reinforcement systems can explain the coexistence of gregariousness and monogamy.

DOI: https://doi.org/10.7554/eLife.25819.001

*For correspondence:
kt66@hunter.cuny.edu

**Competing interests:** The authors declare that no competing interests exist.

## Introduction

Many species of highly gregarious and colonial birds form long-term monogamous pairs (*Goodson et al., 2012*; *Goodson and Kingsbury, 2011*; *Griffith et al., 2010*; *Zann, 1994*). Pair bonding and flocking behaviors are regulated by neuropeptides and dopaminergic reward system (*Goodson et al., 2012*; *Goodson and Kingsbury, 2011*). However, for an animal to be highly social and at the same time monogamous, it must possess two distinct reinforcement systems: one with low selectivity for social stimuli to promote aggregation, and another highly selective for sexual stimuli to promote monogamy. But many communicative stimuli, including birdsong, may serve both social and sexual functions. In such cases, reinforcement may depend on stimulus context: for example, in many solitary songbird males, producing the same song may either attract females or repel rival males (*Kroodsma and Byers, 1991*; *Slater, 2003*). In social songbirds, however, many females and males live in close proximity, which gives females immediate access to numerous males whose songs may sexually attract them. What is it, then, that allows gregariousness and monogamy to coexist? We investigated this question in zebra finches, which are highly social, yet monogamous songbirds (*Griffith et al., 2010*; *Zann, 1994*). Male zebra finches produce a single stereotyped song that can be female-directed or undirected (*Jarvis et al., 1998*; *Scharff and Nottebohm, 1991*;

**eLife digest** While monogamy is rare within the animal kingdom, some species – including humans and many birds – can be highly social and yet sustain monogamous relationships. Zebra finches, for example, are among a number of species of songbirds in which numerous males and females live closely together but maintain monogamous partnerships. Male songbirds use their songs to attract females, who do not themselves sing. But if female birds are attracted to any male song, how do they manage to remain monogamous when surrounded by potential suitors?

In songbirds, and in humans too, a region of the brain called the striatum regulates both social and sexual behaviors. It does this by modulating the release of a molecule called dopamine, which is the brain's reward signal. Tokarev et al. show that hearing songs triggers dopamine release within the striatum of unattached male zebra finches, but has no such effect in single females. Unattached male songbirds will also put up with irritating puffs of air in exchange for being able to watch videos of singing birds, whereas unattached females will not. Female songbirds with partners will tolerate the air puffs, but only if the videos are accompanied with the songs of their own mate.

These findings suggest that song serves as a generic social stimulus for zebra finch males, helping large numbers of birds to live together. By contrast, for a female zebra finch, the song of her partner is a highly selective sexual stimulus. These sex-specific responses to the same socially-relevant stimuli may explain how gregarious animals are able to maintain monogamous pair bonds. More generally, these results are a step towards understanding how brain reward systems regulate social interactions. Studying these mechanisms in songbird species with different social and mating systems could ultimately provide insights into social and sexual behavior in people.

DOI: https://doi.org/10.7554/eLife.25819.002

*Sossinka and Böhner, 1980*; *ten Cate, 1985*; *Woolley and Doupe, 2008*). Males typically tolerate the singing behavior of their neighbors even when housed in crowed cages, although the song is occasionally used in an aggressive context too (*Ihle et al., 2015*). Female zebra finches are attracted to male songs (*Holveck and Riebel, 2007*), but do not sing (*Nottebohm and Arnold, 1976*).

The zebra finch striatal dopaminergic reward circuitry is activated in both social and sexual context (*Banerjee et al., 2013*; *Ihle et al., 2015*; *Iwasaki et al., 2014*; *Sasaki et al., 2006*). In general, there are more dopamine-producing neurons in social than in territorial songbirds (*Goodson et al., 2009*). In zebra finches, gregariousness is correlated with the level of activity in dopaminergic neurons (*Kelly and Goodson, 2015*). Striatal dopamine increases in social situations, e.g., when adult males interact with females (*Ihle et al., 2015*; *Sasaki et al., 2006*), or juvenile males with adult male tutors, and importantly, even without singing in either of these contexts (*Ihle et al., 2015*). During pair formation striatal dopamine levels increase in both sexes (*Banerjee et al., 2013*; *Iwasaki et al., 2014*). In the context of song learning, striatal dopaminergic input is modulated during singing (*Gadagkar et al., 2016*; *Hoffmann et al., 2016*; *Simonyan et al., 2012*). However, although song is an important sexual stimulus in songbirds (*Kroodsma and Byers, 1991*; *Slater, 2003*), there is no direct evidence that hearing songs may affect striatal dopamine in either sexual or affiliative (*Hausberger et al., 1995*) context. Here we performed in vivo imaging and behavioral experiments that show the forebrain dopaminergic system response to song stimulation in zebra finches across sexes and breeding states, in order to distinguish between social and sexual components of song reinforcement in social songbirds.

We developed two complementary experimental approaches. First, we used a delayed positron emission tomography (PET) procedure (*Patel et al., 2008*) in order to measure dopamine neurotransmission (*Laruelle, 2000*) in awake and unrestrained birds. Zebra finches were injected with [$^{11}$C] raclopride radiotracer, which binds to dopamine type 2 (D2) receptors. Instead of acquiring PET immediately, we first stimulated them with song playbacks for 20 min while awake and behaving and scanned them just after the stimulation under general anesthesia (delayed PET, *Figure 1*, see protocol in Materials and methods). Second, we developed an apparatus for assessing song reinforcement behaviorally. This approach is a variant on drug addiction experiments, which typically measure how much rodents are willing to work, or exchange mild punishment, in return for access to dopaminergic stimulants such as cocaine (*Shaham et al., 2000*) (*Figure 2*). We used a song stimulus instead of

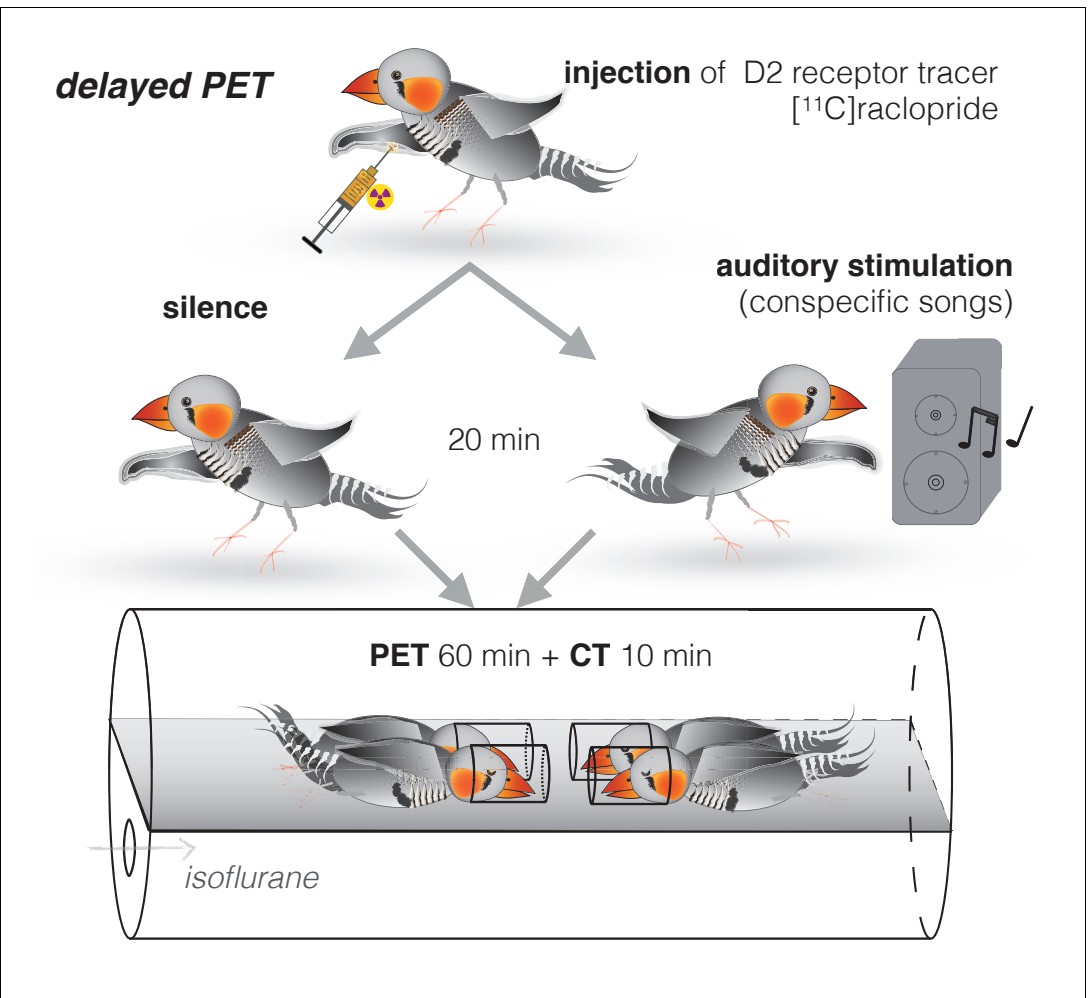

**Figure 1.** Delayed PET of dopamine neurotransmission in response to song stimuli. Adult zebra finches were injected with the D2 receptor tracer [¹¹C]raclopride. Immediately after the injection, birds were either kept for 20 min in quiet conditions or exposed to novel conspecific songs. Each bird was tested in both conditions. PET scan was performed immediately afterwards, in groups of four.
DOI: https://doi.org/10.7554/eLife.25819.003

the drug and measured the extent to which birds were willing to receive mildly aversive air puffs (*Tokarev and Tchernichovski, 2014*) in exchange for hearing song playbacks. Finally, in order to test for causality between dopamine neurotransmission and song reinforcement behavior, we blocked dopamine neurotransmission with a selective antagonist of D2 receptors L-741,626 (*Li et al., 2010*; *Watson et al., 2012*). We used PET to determine the localization of dopaminergic blockage, and then tested behaviorally if blocking of dopamine D2 receptors was sufficient to diminish reinforcing effect of songs.

## Results

We first tested if our delayed PET technique could detect changes in striatal dopamine neurotransmission after hearing song playbacks. We scanned eight unmated female zebra finches, where we expected to find higher levels of dopamine neurotransmission after song playbacks (i.e., lower levels of [¹¹C]raclopride binding), and eight unmated males, where we expected to find a weaker effect, if any. Each bird was scanned twice: after stimulation with a variety of unfamiliar songs (both female-directed and undirected) over 20 min, and after silence over the same duration (*Figure 1*). As expected from the distribution of dopamine receptors in the songbird brain (*Kubikova et al., 2010*),

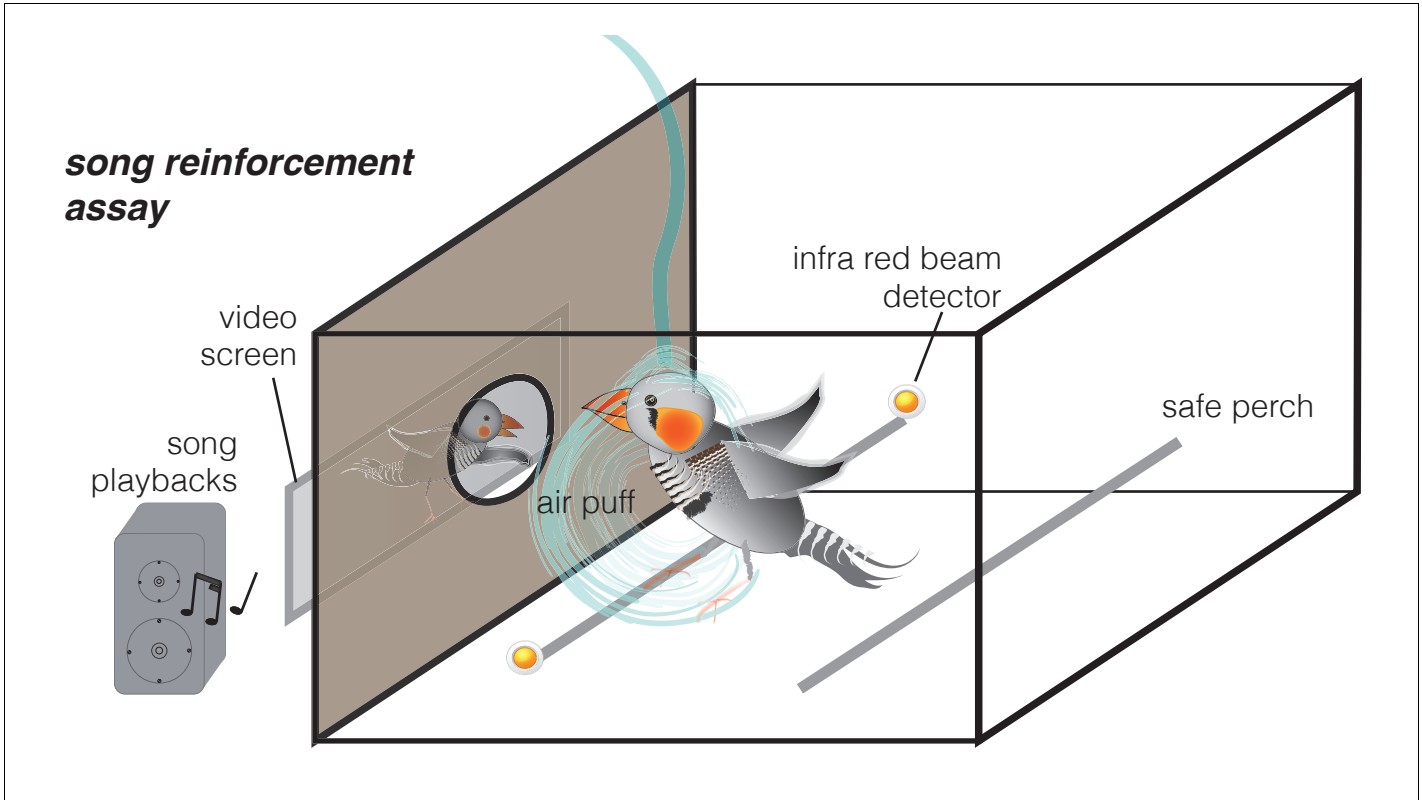

**Figure 2.** Song reinforcement assay. An apparatus for testing the amount of aversive air puffs birds were willing to receive in exchange for hearing songs. Birds voluntarily perched next to a window through which they could see a video of a singing bird. Videos were presented either silently (control) or accompanied with song playbacks. When the infrared beam detected the bird perching next to the window, aversive air puffs were delivered in random (unpredictable) intervals (with a likelihood of 12.5% s).

DOI: https://doi.org/10.7554/eLife.25819.004

the averaged PET map showed that the striatum was the major site of [$^{11}$C]raclopride binding in both conditions in males (*Figure 3a*) and in females (*Figure 3b*). However, against our expectations, lower level of [$^{11}$C]raclopride binding after hearing songs (suggesting increased striatal dopamine neurotransmission) was detected only in the male group. In males, the *song minus silence* parametric difference map showed that song stimulation resulted in significantly lower level of [$^{11}$C]raclopride binding in a part of the striatum (*Figure 3c*; cluster-level $p_{corrected}$ = 0.024, paired t-test corrected for multiple comparisons). Exploratory analysis of individual changes (within the cluster of significant change) showed that [$^{11}$C]raclopride binding was at lower levels in all males after hearing songs by 29 ± 8% (mean ± s.e.m. hereafter; *Figure 3d*; p=0.015, pair-wise t-test). These results, based on PET of D2 receptors, are comparable to the 26.5 ± 8.4% increase in dopamine detected with microdialysis in a study where male zebra finches were presented with females (*Ihle et al., 2015*), confirming that [$^{11}$C]raclopride binding at D2 receptors is a robust indicator of the overall striatal dopamine neurotransmission.

Surprisingly, females lacked any brain areas with significant change in [$^{11}$C]raclopride binding in response to song playbacks. Nevertheless, we produced a mask image from the cluster of significant change in males (*Figure 3c*) and used it as a volume of interest to assess for a possible effect in females. Exploratory analysis of individual changes in females showed no apparent change in striatal [$^{11}$C]raclopride binding in response to song playbacks (*Figure 3e*; 0.4 ± 6%, p=0.737, pair-wise t-test). A direct comparison between males and females showed statistically significant differences in striatal [$^{11}$C]raclopride binding after hearing songs (*Figure 3—figure supplement 1*; p=0.015, t-test). Note, however, that the difference in the magnitude of change between males and females is, at least partially, driven by the low baseline (silence) [$^{11}$C]raclopride binding in females (*Figure 3e*).

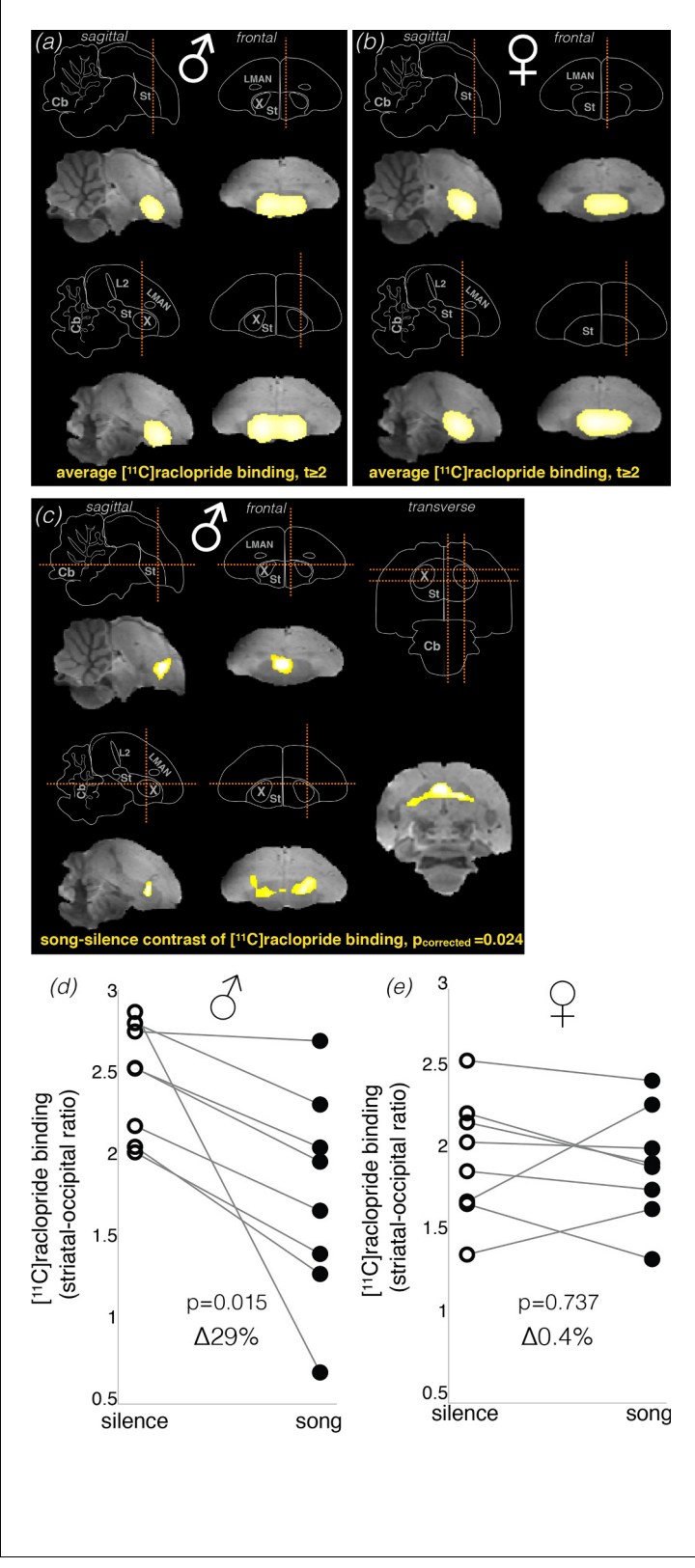

**Figure 3.** Dopamine neurotransmission in response to song stimuli in unmated males and females. Brain schemas in (**a–c**) show: cerebellum (Cb), auditory field L (L2), striatum (St), and song control nuclei Area X (X) and lateral magnocellular nucleus of the anterior nidopallium (LMAN). Section planes are shown as dashed orange lines. (a & b) Bright yellow areas represent the Statistical Parametric Map (SPM, intensity threshold at t ≥ 2) for averaged
*Figure 3 continued on next page*

*Figure 3 continued*

[$^{11}$C]raclopride binding potential in males (**a**) and females (**b**) (n = 8 in both groups). SPM is shown over the brain template magnetic-resonance image. In both males and females [$^{11}$C]raclopride binding was restricted to the striatum. (**c**) SPM of the difference in dopamine neurotransmission as detected by [$^{11}$C]raclopride binding in song and silence conditions in males. SPM reveals significantly lower level of [$^{11}$C]raclopride binding in response to hearing novel conspecific songs in males (pair-wise t statistic, cluster-level $p_{corrected}$ = 0.024), which indicates higher dopamine neurotransmission in this condition. Significant difference was detected in one cluster within the dorsal striatum, mostly outside Area X. (**d**) Analysis of individual changes in [$^{11}$C]raclopride binding in males, comparing song vs. silence. (**e**) Same for females. As no significant cluster was found in females, males' cluster was used as a mask to produce individual values of [$^{11}$C]raclopride binding within the same area.
DOI: https://doi.org/10.7554/eLife.25819.005

The following figure supplements are available for figure 3:

**Figure supplement 1.** Change in dopamine neurotransmission after song playbacks in males and females.
DOI: https://doi.org/10.7554/eLife.25819.006
**Figure supplement 2.** Body and head movements during song playbacks or silence in males and females.
DOI: https://doi.org/10.7554/eLife.25819.007
**Figure supplement 3.** Statistic parametric map (SPM) of differential striatal [$^{11}$C]raclopride binding in male zebra finches at increased threshold.
DOI: https://doi.org/10.7554/eLife.25819.008

The sexually dimorphic striatal response to songs could reflect behavioral or anatomical differences between sexes not related to reinforcement. First, as striatal dopamine neurotransmission correlates with movement (*Cousins and Salamone, 1996*; *Gadagkar et al., 2016*; *Howe and Dombeck, 2016*), we tested if birds tended to move more when hearing song playbacks, in a manner that could explain our results. We analyzed movement in eight males and eight females, in similar conditions to those in our experiments before PET scan: injection of raclopride followed by 20 min of silence or song playbacks. We observed very little of such body movements as flying, hopping and wing-whirring, and also quantitatively tracked the whole body movement (analyzed every 0.3 s for the center of body mass), but there were no significant differences between conditions or sexes (*Figure 3—figure supplement 2*; *Table 1*). Tracking head movement, we observed a significant trend to move the head more during song playbacks in most birds (*Figure 3—figure supplement 2*). However, there was no significant difference between males and females in this respect (*Table 2*). Therefore, mere movement is unlikely to explain our finding of male-specific dopamine response to songs.

Another concern is that our results could simply reflect anatomical dimorphism in the basal ganglia pathway of the premotor song system: in particular, Area X, which has high density of dopamine D2 receptors (*Kubikova et al., 2010*) and receives dopamine during female-directed singing (*Sasaki et al., 2006*), exists only in zebra finch males. However, Area X was mostly excluded from the cluster of significant change (*Figure 3c* and *Figure 3—figure supplement 3*), suggesting that its

**Table 1.** Results of statistical tests to address the differences in body movement in zebra finch males and females in different conditions: in silence or during conspecific song playbacks.
Average Euclidian distance every 0.3 s was measured in the videos for the center of body mass. Bold-face numbers indicate significance levels p≤0.05.

| Box's Test of Equality of Covariance Matrices | Box's M | F | df1 | df2 | p-value |
|---|---|---|---|---|---|
| | 13.334 | 3.756 | 3 | 35280 | **0.01** |
| **Multivariate Tests** (Pillai's Trace) | value | | F | | p-value |
| body movement | 0.175 | | 2.968 | | 0.107 |
| body movement * sex | 0.02 | | 0.21 | | 0.886 |
| **Tests of Between-Subjects Effects** | df | | F | | p-value |
| sex | 1 | | 0.249 | | 0.626 |

DOI: https://doi.org/10.7554/eLife.25819.009

**Table 2.** Results of statistical tests to address the differences in head movement in zebra finch males and females in different conditions: in silence or during conspecific song playbacks.

Average Euclidian distance every 0.3 s was measured in the videos for the position of the beak. Bold-face numbers indicate significance levels p≤0.05.

| Box's Test of Equality of Covariance Matrices | Box's M | F | df1 | df2 | p-value |
|---|---|---|---|---|---|
| | 4.004 | 1.128 | 3 | 35280 | 0.336 |
| **Multivariate tests** (Pillai's Trace) | Value | | F | | p-value |
| head movement | 0.348 | | 7.468 | | **0.016** |
| head movement * sex | 0.016 | | 0.225 | | 0.643 |
| Tests of Between-Subjects Effects | df | | F | | p-value |
| sex | 1 | | 0.598 | | 0.454 |

DOI: https://doi.org/10.7554/eLife.25819.010

contribution was small, if any. This is in line with the finding that Area X does not respond to auditory stimulation in awake songbirds, except for error signals during singing (*Gadagkar et al., 2016*).

Given that the expectation of reward is only one of several scenarios that could explain the unanticipated pattern of striatal dopamine neurotransmission that we observed (*Cousins and Salamone, 1996*; *Gadagkar et al., 2016*; *Hoffmann et al., 2016*; *Howe and Dombeck, 2016*; *Kubikova and Kostál, 2010*; *Riters, 2011*; *Salimpoor et al., 2011*; *Schultz, 2002*; *Stuber et al., 2008*), we developed an independent method for assessing the effect of song reinforcement in male and female zebra finches. In order to directly estimate song reinforcement we paired the song stimulus with a mild punishment. We presented the same birds that had been scanned earlier for dopamine with video of a perching male (*Figure 2*). Each bird was presented with two daily sessions of videos over ten days (20 sessions, 20 min each). In ten sessions the video was played in silence, and in the alternating ten sessions, it was accompanied by song playbacks (the same mix of initially unfamiliar songs as in the PET experiments). When a bird perched next to the window facing the video display, it would occasionally receive a mildly aversive air puff, in random intervals and without warning. We assessed reinforcement by measuring the number of air puffs the bird was willing to tolerate in return for the stimulus, comparing the silent sessions to the song playback sessions.

We found that males voluntarily received many more air puffs during song playback sessions compared to silent sessions (*Figure 4*; p=0.001, paired t-test); they appeared attentive during the sessions but did not show any aggressive behavior. Females, on the other hand, showed little motivation to hear song playbacks: their tendencies to receive air-puffs were moderate and did not differ significantly across song playback and silent sessions (*Figure 4*; p=0.267, paired t-test).

To test whether the song reinforcement we observed in males was dependent on dopamine neurotransmission, we used the D2 receptor antagonist L-741,626 to interfere with D2 receptors. First, we performed a whole brain PET after injections of L-741,626 in order to determine the localization of dopaminergic blockage: as expected, changes in [11C]raclopride binding were observed exclusively in the striatum (*Figure 5*). We found substantially lower levels of the striatal binding of [11C] raclopride after L-741,626 injection compared to saline (*Figure 5—figure supplement 1*). Therefore, L-741,626 blocks D2 receptors in the songbird striatum as it does in rodents (*Li et al., 2010*; *Watson et al., 2012*) and primates (*Achat-Mendes et al., 2010*). We then tested song reinforcement in four males with our air-puff apparatus as described before, but after injections of either L-741,626 or saline on alternate sessions. On the days of L-741,626 injections, the animals were still active and approached the video, but stimulation with song playbacks no longer increased the number of air puffs they were willing to receive, while on the alternate days of saline injections, song reinforcement was similar to that of untreated males (*Figure 6*; see *Table 3* for statistics).

How is it that song stimuli are reinforcing in unmated males but not in unmated females? We hypothesized that the non-selective dopamine neurotransmission by unfamiliar songs in males might reflect a social function, but in females, song reinforcement might be exclusively sexually driven, as a part of the mate choice (*Riebel, 2009*). A possible explanation to those counterintuitive results is that reinforcement could be much more selective in females. We therefore measured song reinforcement in six mated females that were ready to breed (*Figure 7—figure supplement 1*). We

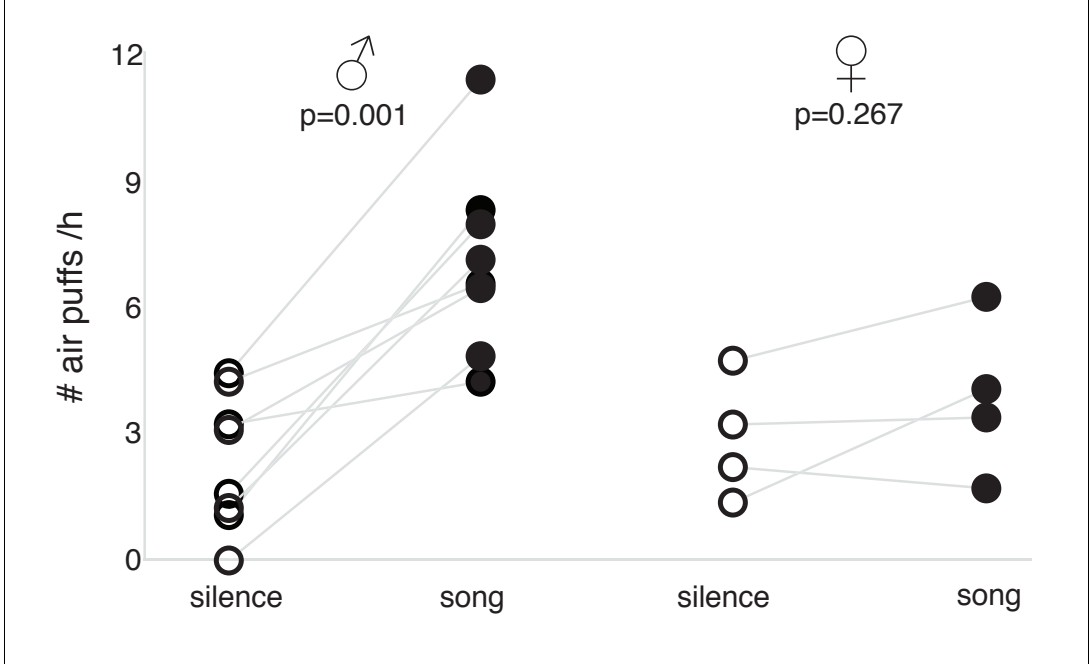

**Figure 4.** Song reinforcement in unmated males and females. Rate of air puffs (per hour) birds received during song playback and silent sessions: in males (*left*) and females (*right*) (n = 8 and n = 4, respectively; p-values for paired t-test shown).

DOI: https://doi.org/10.7554/eLife.25819.011

...:companied with the songs of their mates, ...ed males, and video alone. The mated ...otivation to tolerate air puffs in return to ...ceive many air puffs in return for hearing

...ern of striatal dopamine neurotransmission ...ed PET, we compared two sets of stimuli: ...ced by other mated males (in both condi-...emale-directed and undirected songs). We detected a cluster of vox-...inding in response to mate song in a small part of the medial dorsal striatum (*Figure 8a,b*), however, the difference across those voxels did not survive correction for multiple comparisons (*Figure 8b*). An exploratory post-hoc analysis of individual differences in the same area found that [$^{11}$C]raclopride binding was 12 ± 4% lower in response to mate song compared to non-mate song (*Figure 8c*; p=0.042, paired t-test). These differences suggested a weak trend for higher levels of dopamine transmission in response to mates' songs in females, but this borderline effect should be treate... ...ture studies.

## Discussion

...ern of sexual dimorphism in dopaminergic ...sulted in higher levels of striatal dopamine ...viorally too, unfamiliar song playbacks were ...e D2 receptors extinguished song reinforce-...gic reward system. In unmated females, hear-...ion, and playbacks were not reinforcing ...rongly reinforcing, with high specificity, but ...eurotransmission in response to mate song ...tal dopamine neurotransmission and behav-...ld and non-specific positive reinforcement.

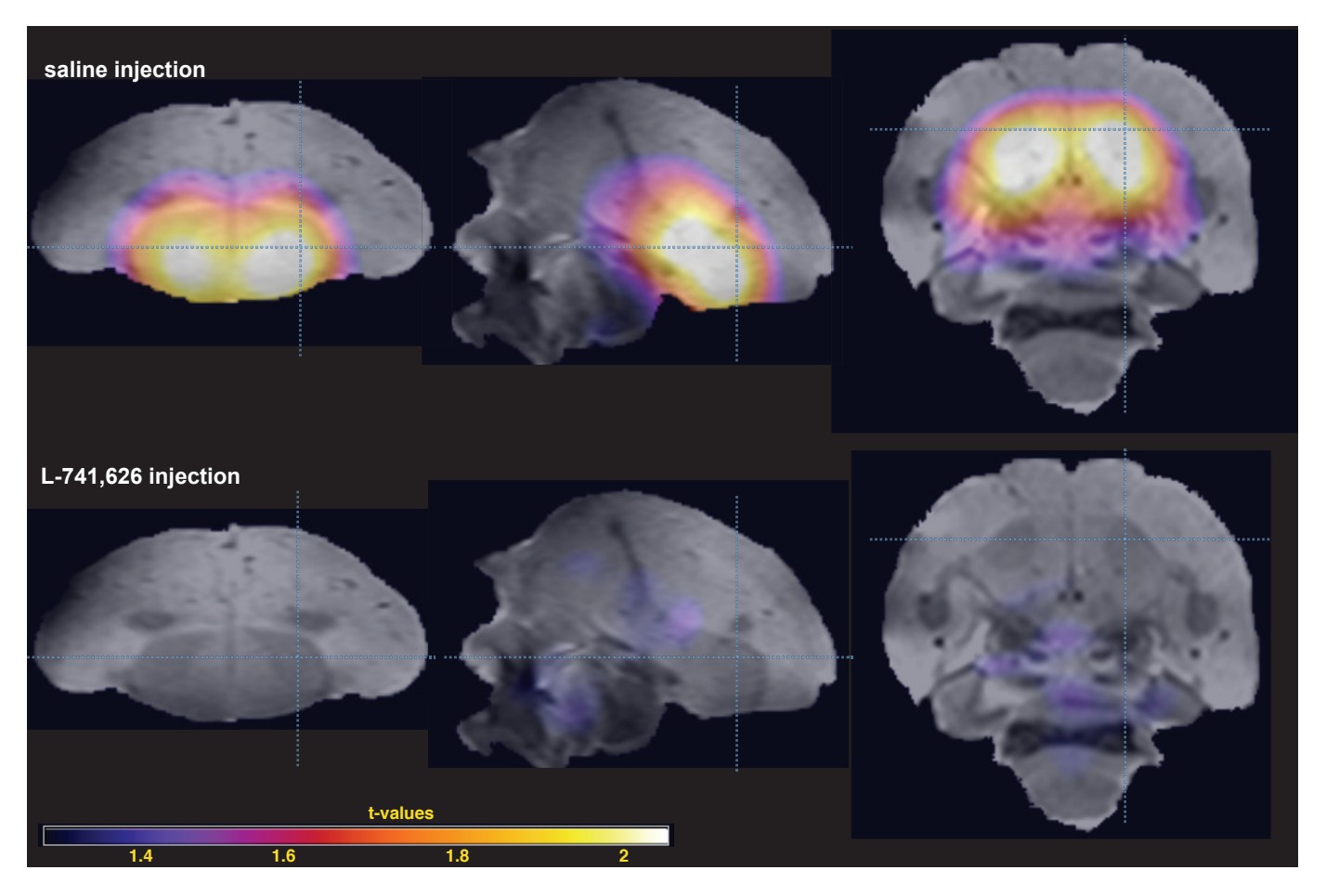

**Figure 5.** Blockage of D2 receptor binding activity by L-741,626. Statistical parametric map of average [$^{11}$C]raclopride binding is shown over the zebra finch brain template magnetic-resonance image: after saline injection (*top*) and L-741,626 injection (*bottom*) (n = 2 in both conditions; t-values on the insert). Sagittal (*left*), frontal (*middle*) and transverse (*right*) sections are shown; dashed light-blue lines show section planes.
DOI: https://doi.org/10.7554/eLife.25819.012

The following figure supplement is available for figure 5:

**Figure supplement 1.** L-741,626 activity at the striatal D2 receptors in zebra finches.
DOI: https://doi.org/10.7554/eLife.25819.013

This is consistent with a social, perhaps affiliative function of birdsong to promote aggregation (*Hausberger et al., 1995*). In females, both behavioral and dopaminergic responses to song were high-threshold and mate-selective, consistent with a sexual function to promote monogamy. However, even though behaviorally mated females showed strong reinforcement to mate song, their striatal dopaminergic responses to mate song were barely detectable. This discrepancy will require further assessment in future studies. Note that there are several open questions about the receptor mechanisms that could account for the sexual dimorphism we observed, including different receptors expression levels, different densities of dopaminergic cells, different reuptake mechanisms and different ratios of D1/D2 receptors. For example, it should be tested whether D1 receptors, which are known to be important for reinforcement (*Robbins and Everitt, 1996*), are also crucial in the reward mechanism of song in zebra finches.

A simple evolutionary scenario can explain the pattern of sexual dimorphism we observed. Territorial songbird males respond aggressively to intruders and are easy to irritate with conspecific song playbacks (*Kroodsma and Byers, 1991*; *Slater, 2003*). Females may show strong preference to certain male song features but are generally attracted to conspecific songs (*Kroodsma and Byers, 1991*; *Slater, 2003*). Monogamy could be sustained during an evolutionary transition from the

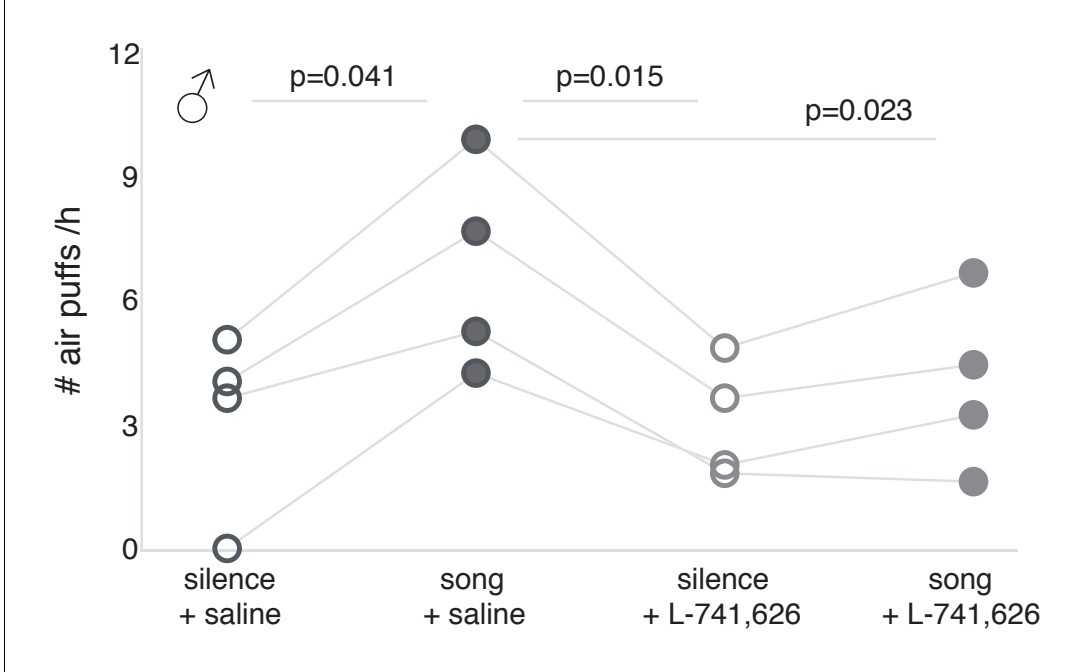

**Figure 6.** Song reinforcement after dopamine receptor blockage. Number of air puffs (per hour) birds received during silent and song playback sessions: after saline injection (*left*); after L-741,626 injection (*right*) (n = 4; significant p-values are shown for general linear model for repeated measurements; see *Table 3* for statistics).

DOI: https://doi.org/10.7554/eLife.25819.014

territorial to gregarious behavior if male evolved high tolerance to song while female simultaneously co-evolved highly selective reinforcement threshold to songs. Our results are consistent with such a scenario. Future studies could test this hypothesis further by systematic examination of sexual dimorphism across territorial and social species of songbirds, and in species where both sexes sing. We would expect to see a lack of song reinforcement in non-social territorial songbirds, at least outside the breeding period. But possibly, aggressive reaction might also increase brain dopamine, and one

**Table 3.** Results of statistical tests to address the differences in tolerance to air puffs in zebra finch males in different conditions: in silence or during conspecific song playbacks after saline injections, or same after injection of dopamine receptor antagonist L-741,626.
Bold-face numbers indicate significance levels $p \leq 0.05$.

| Mauchly's Test of Sphericity | Mauchly's W | df | $\chi^2$ | p-value |
|---|---|---|---|---|
| # air puffs/h | 0.022 | 5 | 6.604 | 0.318 |

| Tests of Within-Subjects effects | df | F | p-value |
|---|---|---|---|
| # air puffs/h | 3 | 7.96 | **0.007** |

| pair-wise post-hoc LSD tests | p-value |
|---|---|
| song + saline vs silence + saline | **0.041** |
| song + saline vs silence + L-741,626 | **0.015** |
| song + saline vs song + L-741,626 | **0.023** |
| silence + saline vs silence + L-741,626 | 0.814 |
| silence + saline vs song + L-741,626 | 0.394 |
| song + L-741,626 vs silence + L-741,626 | 0.122 |

DOI: https://doi.org/10.7554/eLife.25819.015

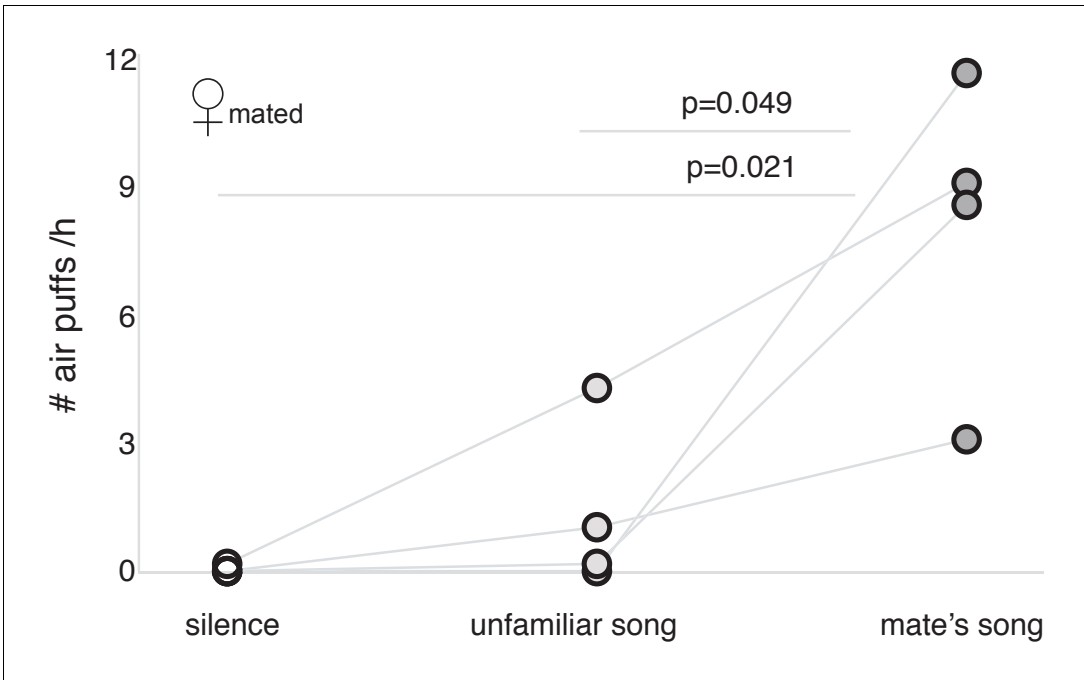

**Figure 7.** Song reinforcement in mated females. Number of air puffs (per hour) mated females received in exchange for silence, non-mate song (from male mated with another female), and mate's song (n = 4; significant p-values are shown for general linear model for repeated measurements; see *Table 4* for statistics).
DOI: https://doi.org/10.7554/eLife.25819.016

The following figure supplement is available for figure 7:

**Figure supplement 1.** Experimental procedures for measuring dopamine neurotransmission in female zebra finches in response to their mates' songs.
DOI: https://doi.org/10.7554/eLife.25819.017

should try to carefully dissect such effects. For example, it was shown that fighting cocks (***Thompson, 1964***) and Siamese fighting fish (***Thompson, 1963***) may perceive seeing a potential opponent as a reinforcing stimulus; so, either they may look forward to the fight, or it is an anticipation of reward after winning the fight. In Siamese fighting fish, it was shown that dominant males are more likely to use such stimuli than subordinate (***Baenninger, 1970***). Avian species demonstrate a wide range of social structures, so the reinforcement value of social clues may vary greatly among them.

**Table 4.** Results of statistical tests to address the differences in tolerance to air puffs in mated zebra finch females in different conditions: in silence and during playbacks of songs of unfamiliar males or their mates.
Bold-face numbers indicate significance levels p≤0.05.

| Mauchly's Test of Sphericity | Mauchly's W | df | $\chi^2$ | p-value |
|---|---|---|---|---|
| # air puffs/h | 0.484 | 2 | 1.453 | 0.484 |

| Tests of Within-Subjects Effects | df | F | p-value |
|---|---|---|---|
| # air puffs/h | 2 | 13.139 | **0.006** |

| pair-wise post-hoc LSD tests | p-value |
|---|---|
| mate's song vs silence | **0.021** |
| mate's song vs non-mate song | **0.049** |
| non-mate song vs silence | 0.259 |

DOI: https://doi.org/10.7554/eLife.25819.018

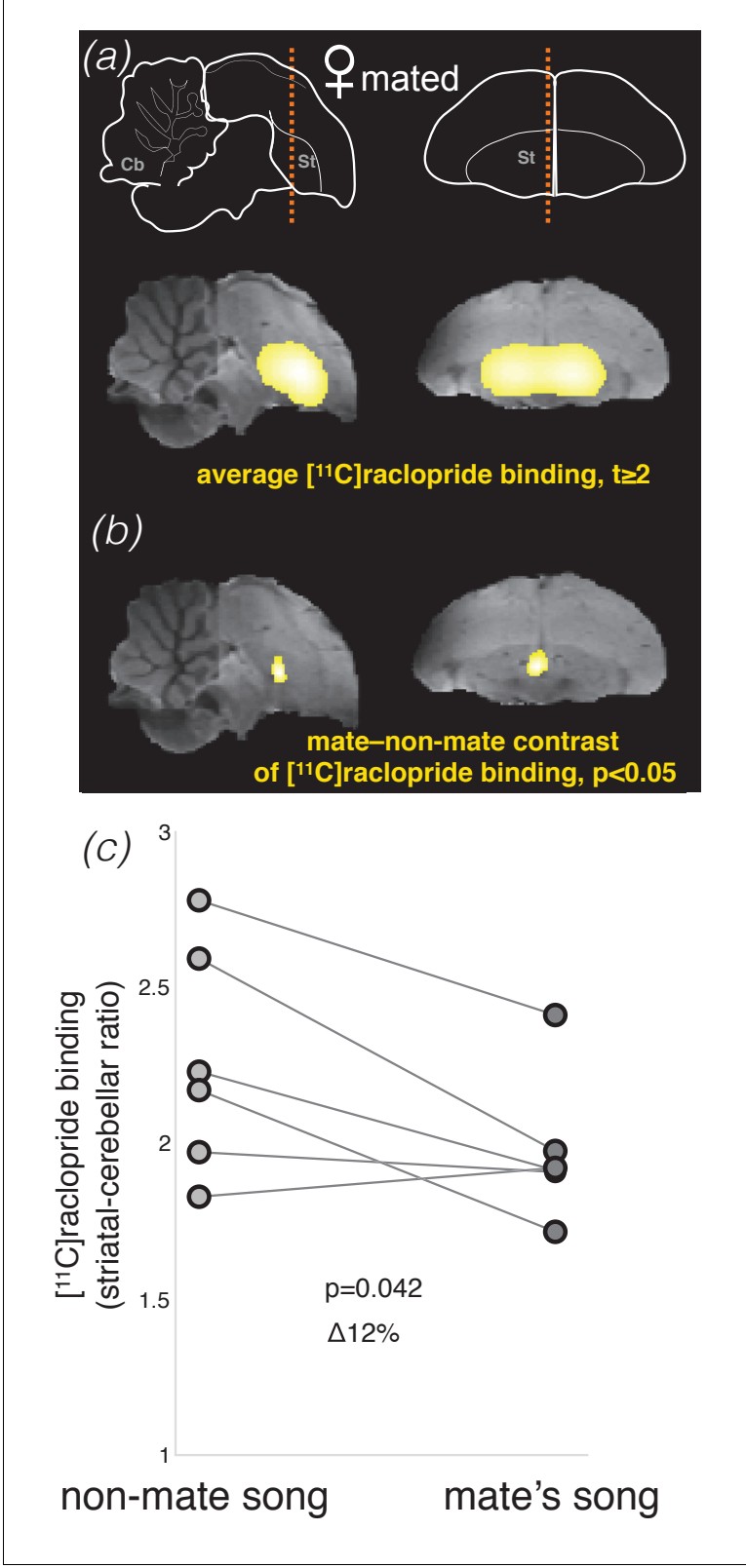

**Figure 8.** Dopamine neurotransmission in response to song stimuli in mated females. (a) Brain schemas as in *Figure 3a,b*. Statistical parametric map (SPM, intensity threshold at t ≥ 2) for average [11C]raclopride binding is shown over the zebra finch brain template magnetic-resonance image. (b) SPM of the difference in [11C]raclopride binding in response to non-mate song and mate's song in mated females (n = 6; pair-wise t statistic, p<0.05). This *Figure 8 continued on next page*

*Figure 8 continued*
cluster, however, did not survive correction for multiple comparisons ($p_{corrected}$ = 0.6, paired t-test corrected for multiple comparisons). (c) Individual changes in [$^{11}$C]raclopride binding in this insignificant cluster in mated females, non-mate song vs. mate's song. Supplementary information.
DOI: https://doi.org/10.7554/eLife.25819.019

In sum, a sexually dimorphic activation of the dopaminergic reward circuitry that we observed in our study could provide a joint mechanism for aggregation and pair-bonding, two seemingly conflicting characteristics of the social structure of zebra finches and other gregarious yet monogamous species.

## Materials and methods

### Experimental design

This study was conducted in accordance with the guidelines of the US National Institutes of Health and was approved by the Institutional Animal Care and Use Committees of Hunter College of the City University of New York (protocol 'OT imaging 10/18–01') and Weill Cornell Medical College (protocol #2010–0003).

Eleven adult male and seventeen adult female zebra finches (*Taeniopygia guttata)* bred at Hunter College (room temperature 19–24˚C, 12:12 hr light/dark cycle) were used in the neuroimaging experiments. The animals were raised by both parents until adulthood and spent their life, except for the time of experiments, in the colony room with possibility to engage in social interactions with other zebra finches. All males and nine of the females were non-mated, eight other females were mated in breeding pairs.

The concept of our work was similar to a human study, where favorite musical pieces were shown to increase striatal dopamine levels (*Salimpoor et al., 2011*), but we employed a modification in PET protocol that allowed to obtain measurements that reflected changes in dopamine release in awake songbirds. Before imaging, the non-mated animals were injected [$^{11}$C]raclopride and then either exposed to recorded songs of unfamiliar male zebra finches or kept in quiet conditions for 20 min (*Figure 1*). This time interval was chosen according to the $^{11}$C half-life of 20 min and its detectability with the current PET technique. PET and anatomical X-ray computed tomography (CT) images were acquired immediately afterwards using an Inveon Research Workplace (Siemens). Delayed PET scans for dopamine are well established in several animal species (*Marzluff et al., 2012*; *Patel et al., 2008*), but since this is a novel method for measuring striatal responses to birdsong, we describe it in detail as a protocol in the next section.

Eight mated female zebra finches were tested in a similar experiment, but with songs of either their own mate or another mated male; they were also synchronized in their breeding cycle so that during stimulation and PET they would be in similar hormonal states (*Figure 7—figure supplement 1*. The females were kept together with their mates for the first week after hatching of the offspring but then were moved (together with offspring) to the nursery room in the absence of adult males until post-hatch day 30, after which they would reunite with their mates. This cycle is routinely performed in the laboratory to produce juvenile zebra finches not exposed to adult male song, which we use in other studies. For this experiment, we used females that had gone through this cycle several times, and stimulation/scanning took place shortly before their return to the mates (*Figure 7—figure supplement 1*). Scanning procedures were the same as in the previous experiment and are described in more detail in the next section.

Eight of the males, four unmated females and four of the mated females were also tested in a behavioral paradigm for preferences to the auditory stimuli that had been used in the PET experiments (*Figure 2*). We modified our socially-reinforced auditory discrimination paradigm (*Tokarev and Tchernichovski, 2014*), so that after a period of isolation the zebra finches were attracted to a video of a male (*Ljubičić et al., 2016*). The video was played either in silence (20 min) or with the same auditory stimuli as in the PET experiments: a mix of songs of unfamiliar male zebra finches for the males and unmated females, and songs of unfamiliar males or mates for the mated females (20 min). The order of auditory accompaniment (silence/songs) in each session was random;

each animal was tested in 10 sessions. In order to see the video and be closer to source of auditory stimulation, the animals had to sit on a perch that produced air puff in a random manner controlled by Bird Puffer software (http://soundanalysispro.com/bird-puffer). We previously determined that random air puffs with a probability of ~2/minute are well tolerated by the birds. Our software automatically registered the bird's perching activity, delivered the air puffs, and kept continuous records of air puffs that each bird received. We then analyzed during which stimulation the animals were willing to receive more air puffs.

We also tested whether the movement might account for observed differences in striatal dopamine release. If dopamine level changes were due to movement, then movement should differ across treatments: higher in zebra finch males but not females when hearing songs compared to when they are kept in silence. To test if this were the case, we performed an additional control experiment with a new group of 8 males and eight females, where we simulated the song vs. silence pre-PET conditions (including transfer to the same room and raclopride injection), and also video tracked birds' movement. We monitored for such body movements as flying, hopping and wing-whirring, as well as quantitatively analyzed videos for Euclidian distances every 0.3 s for the center of body mass and beak to continuously track changes in position of body and head, respectively.

## Injections of L-741,626

To detect whether dopamine neurotransmission was necessary for the observed behavioral effects in males, four of them were injected with L-741,626 (Sigma-Aldrich, Saint Louis, MO, USA), a very selective antagonist of D2-receptors, which had been used to study the function of D2-receptors in rodents (*Dai et al., 2016*; *Li et al., 2010*; *Watson et al., 2012*) and primates (*Achat-Mendes et al., 2010*). We injected L-741,626 intraperitoneally at 3.33 µg/g body weight, within the range described for rodents (*Li et al., 2010*; *Watson et al., 2012*), diluted in saline (acetic acid was added to increase solubility at first, then pH was neutralized by caustic soda solution). The L-741,626 injections were administered 30 min before each test with at least 48 hr between treatments, 5 times for each animal, with an intra-individual control of sham injections (saline) of the same volume.

## Simultaneous PET on four zebra finches to measure dopamine released during auditory stimulation in awake unrestrained state

We established a minimally invasive method for in vivo imaging in zebra finches to measure dopamine neurotransmission in four awake unrestrained animals simultaneously; these measurements may be taken multiple times allowing for intra-subject comparisons (*Figure 1*). Due to their small size compared to the available imaging volume of our micro-PET, we were able to scan four birds simultaneously. Thus, the experiments were done in tetrads, with two animals in one condition, and two animals in another, and then the conditions were reversed for them in the subsequent PET scan. [$^{11}$C]raclopride was delivered via intravenous (i.v.; ulnar vein) or intraperitoneal (i.p.) bolus injections that lasted around 1 min or less; radioactivity doses were ~300 µCi or less, in solutions of 150 µl for i.p. injections and 100 µl for i.v. injections with [$^{11}$C]raclopride mass at ~0.3 nmol/g (body weight). Usage of [$^{11}$C]raclopride to track changes in dopamine levels has been validated in studies with simultaneous microdialysis (*Morris et al., 2008*; *Normandin et al., 2012*).

When dopamine is released, decrease in radioactive [$^{11}$C]raclopride signal is mediated through direct competition between these two molecules for D2 receptors (*Fisher et al., 1995*) and as a result of D2 receptors switching from low to high affinity for dopamine but not raclopride (*Fisher et al., 1995*; *Seeman et al., 1994*); also, the striatal [$^{11}$C]raclopride signal does not rebound after its decline once dopamine is released (*Endres et al., 1997*). Therefore, differences in dopamine neurotransmission between zebra finches exposed to song playbacks and silence observed in our work were likely due to experimental conditions, even though imaging was performed after stimulation (*Yoder et al., 2008*). This method of delayed PET (aka 'awake uptake') was first used to detect changes in dopamine levels in freely moving rats (*Patel et al., 2008*). A similar protocol was also used in songbirds (crows), but with [$^{18}$F]−2-fluoro-2-deoxy-D-glucose to detect general brain activation in response to visual stimuli (*Marzluff et al., 2012*).

The animals were let to recover after handling for 1–2 min and then were kept individually either in quiet conditions (20 min) or were presented with recordings of various zebra finch songs (one novel song every 15 s during 20 min), thus providing stimulation almost immediately after

radioligand injection, similarly to previous studies (*Marzluff et al., 2012*; *Patel et al., 2008*). Food and water were provided ad libitum. None of the birds sang or attempted to sing during the 20 min of the experiment (in all conditions). Their behavioral activity was at minimum during the experiment with no drinking or feeding, and only occasional perching. This suggested that the difference in experimental conditions (song playbacks or silence) would be the sole factor in possible differences in dopamine neurotransmission. Immediately after the experiment, the animals were sedated ~2 min under 3% isoflurane in O2, 2 L/min, and transferred into a custom-made plexiglass chamber with 4 head holders made from vinyl tubes; their bodies were additionally fixed with a surgical tape to reduce spontaneous movements during scanning. Animal placement (2 in radial, 2 in axial direction; heads facing towards the center of the imaging volume) was chosen to maximize image quality (*Siepel et al., 2010*). The chamber was then placed in the micro-PET scanner, and anesthesia was reduced to 2% isoflurane. Acquisition of the radioactive signal lasted 60 min and was followed by an anatomical CT scan of 10 min duration. Differences in radioactive signal acquired during the PET scan were expected to reflect dopamine release during auditory stimulation, as after [$^{11}$C]raclopride is displaced by dopamine its level does not rebound within this time frame despite clearance of dopamine and even with continuous infusion of [$^{11}$C]raclopride (*Endres et al., 1997*), while we performed single bolus injection. We were able to inject a sufficient amount of radiotracer to obtain images of [$^{11}$C]raclopride uptake, and all animals recovered quickly after the scan. We established that both i.v. and i.p. injections of [$^{11}$C]raclopride produced a radioactive signal in striatum that was detectable by micro-PET, and the data from birds after i.v. and i.p. injections of [$^{11}$C]raclopride overlapped and therefore were combined. Thus, both injection methods appeared to be effective for detection of dopamine level changes. We recommend i.p. injections for future research, as they are faster and easier to perform, require less handling and thus are less stressful for animals (and experimenters).

We also performed an additional PET scan on four males that had been tested with the D2 receptor antagonist, L-741,626, to confirm that it blocked binding at the receptor. Two of them were injected L-741,626 solution and two others saline 30 min before [$^{11}$C]raclopride injection. The rest of the procedure was the same.

## Radiochemistry

The radiotracer [$^{11}$C]raclopride was synthesized on-site immediately before each experiment at the Citigroup Biomedical Imaging Center, Weill Cornell Medical College, following standard procedures (*Broft et al., 2015*; *Mawlawi et al., 2001*). The average specific activity of [$^{11}$C]raclopride was 6046 mCi/μmol. [$^{11}$C]raclopride was isolated and formulated into an isotonic solution containing 5–7% ethanol, with concentration of 0.13 μg/mL. Although alcohol could potentially influence behavioral state of the animal, the amount injected in our experiments (~0.3 g/kg) was substantially lower than that causing an intoxicated stupor in a previous study (2–3 g/kg) (*Olson et al., 2014*) and importantly was similar across all experimental conditions.

## PET image preparation and statistical analysis

PET imaging data were first processed in PMOD software (http://www.pmod.com). As four animals were scanned simultaneously at each experiment, raw images were separated into four zones around each brain and cropped accordingly in PMOD software. PET data were summed across 6 evenly distributed time points for each scan. Further, PET data were processed and analyzed in SPM12 software (http://www.fil.ion.ucl.ac.uk/spm).

Anatomical CT images were transformed into standardized stereotaxic space and aligned with a 3D magnetic resonance imaging atlas of the zebra finch brain, which also references common brain areas (*Poirier et al., 2008*). All PET images were corrected for volume-to-volume motion by interframe realignment and then co-registered to the subject's anatomical CT image. All alignment transformations were visually inspected to ensure that there was no mismatch with the template brain image. Datasets of three males, one unmated and two mated females were discarded because of difficulties with alignment of the images due to motion during scans. Data from the remaining 22 animals were analyzed further.

[$^{11}$C]raclopride binding potential for dopamine D2 receptors in each voxel was calculated using a simplified reference region method (*Gunn et al., 1997*; *Lammertsma et al., 1996*; *Patel et al.,*

*2008*), with the cerebellum as the reference region, since it does not contain detectable D2 receptors and is traditionally used for determination of nonspecific binding and free radiotracer in the brain (*Lammertsma et al., 1996*; *Litton et al., 1994*): $(C_{St}–C_{Cb})/C_{Cb}$, where $C_{St}$ is radioactivity concentration in striatal (St) voxels (or anywhere else outside the reference region), and $C_{Cb}$ is averaged radioactivity concentration in cerebellum (Cb). Therefore, [$^{11}$C]raclopride binding potential was represented by a striatal-cerebellar ratio (SCR) of radioactive concentrations (*Patel et al., 2008*). As [$^{11}$C]raclopride and dopamine compete for D2-receptors, decrease in [11C]raclopride binding potential indicates an increase of dopamine concentration (*Endres et al., 1997*; *Fisher et al., 1995*) and thus reflects increased dopamine neurotransmission (*Laruelle, 2000*; *Martinez et al., 2003*). Statistical parametric maps of [11C]raclopride binding potential change were produced by comparing the parametric SCR maps of the two scan sessions (song playbacks and quiet condition, or mate's and unfamiliar songs); comparisons between two conditions were performed with paired t-tests, with two-tailed probability value of p<0.05 chosen as statistically significant (*Urban et al., 2012*). Clusters of significant change were identified in xjView (http://www.alivelearn.net/xjview) at p<0.05; p values corrected for multiple comparisons were calculated for each cluster of contiguous voxels at a t threshold of 3.56 within a search volume equal to the whole brain and an effective spatial resolution of 1.4 mm full-width at half maximum (FWHM) (*Salimpoor et al., 2011*). Mean binding potential values were extracted from the significant cluster for each individual, and the normalized percent change in dopamine level was calculated as $\Delta = (SCR_{silence}–SCR_{song}) \times 100/SCR_{silence}$.

## Acknowledgements

This work was funded by National Science Foundation (grants #1261872, 0956306 and 1065678) and National Institutes of Health (grant # DC04722-17). We thank Michael Synan and Yeona Kang for help with PET image processing.

## Additional information

### Funding

| Funder | Grant reference number | Author |
|---|---|---|
| National Science Foundation | 1261872 | Kirill Tokarev<br>Ofer Tchernichovski |
| National Science Foundation | 0956306 | Henning U Voss |
| National Science Foundation | 1065678 | Santosh A Helekar |
| National Institutes of Health | DC04722-17 | Kirill Tokarev<br>Ofer Tchernichovski |

The funders had no role in study design, data collection and interpretation, or the decision to submit the work for publication.

### Author contributions

Kirill Tokarev, Conceptualization, Data curation, Software, Formal analysis, Funding acquisition, Validation, Investigation, Visualization, Methodology, Writing—original draft, Writing—review and editing; Julia Hyland Bruno, Data curation, Investigation, Writing—review and editing; Iva Ljubičić, Data curation, Software, Validation, Methodology, Writing—review and editing; Paresh J Kothari, Resources, Methodology, Writing—review and editing; Santosh A Helekar, Conceptualization, Data curation, Validation, Methodology, Writing—review and editing; Ofer Tchernichovski, Conceptualization, Resources, Data curation, Software, Supervision, Funding acquisition, Validation, Methodology, Writing—original draft, Project administration, Writing—review and editing; Henning U Voss, Conceptualization, Resources, Data curation, Software, Formal analysis, Supervision, Funding acquisition, Validation, Investigation, Visualization, Methodology, Project administration, Writing—review and editing

## Author ORCIDs

Kirill Tokarev http://orcid.org/0000-0002-2129-1324
Paresh J Kothari http://orcid.org/0000-0002-1590-8682

## Ethics

Animal experimentation: This study was conducted in accordance with the guidelines of the US National Institutes of Health and was approved by the Institutional Animal Care and Use Committees of Hunter College of the City University of New York (protocol 'OT imaging 10/18-01') and Weill Cornell Medical College (protocol #2010-0003).

## Decision letter and Author response

Decision letter https://doi.org/10.7554/eLife.25819.022
Author response https://doi.org/10.7554/eLife.25819.023

## Additional files

### Supplementary files

• Transparent reporting form
DOI: https://doi.org/10.7554/eLife.25819.020

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
