## [Decision Letter]

[Editors’ note: this article was originally rejected after discussions between the reviewers, but the authors were invited to resubmit after an appeal against the decision.]

Thank you for submitting your work entitled "Sexual dimorphism in striatal dopaminergic responses promotes monogamy in social songbirds" for consideration by *eLife*. Your article has been reviewed by three peer reviewers, one of whom, Naoshige Uchida (Reviewer 3), is a member of our Board of Reviewing Editors, and the evaluation has been overseen by the Reviewing Editor and a Senior Editor.

Our decision has been reached after consultation between the reviewers. Based on these discussions and the individual reviews below, we regret to inform you that your work will not be considered further for publication in *eLife*.

Tokarev and colleagues used a delayed PET method to test the role of striatal dopamine in social interactions in zebra finch. Specifically, the authors show that song playback increased the PET signal in the striatum only in males but not in females. Further experiments showed that the PET signal in the striatum increases in female birds only when mated females were exposed to the song of their mated males. Behaviorally, unmated males were willing to obtain song stimulation in exchange of mildly aversive air puffs whereas unmated females weren't. The authors found that mated females were willing to obtain the song of their mated males but not that of other males (unfamiliar song). Finally, the authors demonstrate that D2 dopamine receptor blockade impaired the reinforcing property of songs in males. These results are potentially interesting as it demonstrates positively reinforcing properties of song and the involvement of striatal dopamine in this process.

However, the reviewers found some substantive concerns. First, the authors do not provide evidence that the striatal dopamine signals are not related to animals' movement. Second, the interpretation of the signal obtained using the delayed-PET method is not well explained or is rather confusing. During discussion, we have additionally contacted an expert of PET techniques. This person largely agreed the Reviewer 3's concerns. His/her comments are summarized below:

Typically, a parametric pharmacokinetic model is used to derive a measure of non-displaceable [^11^C]raclopride binding potential for each voxel or region of interest. This is used as a proxy for "D2/D3 receptor availability." Because DA competes with [^11^C]raclopride for available D2/D3 receptors, a reduction in [^11^C]raclopride binding potential following a cognitive/behavioral or pharmacological challenge is taken as evidence for DA release. For example, administering amphetamine will cause striatal [^11^C]raclopride binding potential to decrease (compared to administration of placebo). In the few studies that have also obtained simultaneous direct measures of DA release (typically through microdialysis), the magnitude of stimulus-induced striatal DA release is strongly correlated with the size of the reduction in [^11^C]raclopride binding potential after stimulant administration.

With all that said, the pharmacokinetic models are not particularly robust. Practically, this places significant constraints on experimental design. I raise this point because it seems that the authors used a "delayed PET" approach that is highly nonstandard. In human PET imaging, investigators will typically inject [^11^C]raclopride, measure signal until the tracer reaches equilibrium to obtain a pre-challenge binding potential estimate, introduce the cognitive/behavioral/drug challenge, and then measure signal again to get a post-challenge binding potential estimate. It's this binding potential delta (reflecting the degree to which a challenge reduces binding potential) that's then compared between groups or correlated with some individual difference measure. In principle, it's correct to interpret a challenge-induced increase in [^11^C]raclopride binding potential as reflecting a challenge-induced decrease in DA. In practice, though, this is uncommon and such a finding would either have to be supported by corroborative measures or a compelling, a-priori mechanistic prediction. I should say that their design may be entirely appropriate. What concerning is that they don't provide a coherent justification for this non-standard design, nor do they offer any validation that their models fit the data well given this design.

The way the authors talk about PET measurements is incredibly confusing. [^11^C]raclopride PET does not measure "dopamine receptor activity;" it measures [^11^C]raclopride binding potential! A change in [^11^C]raclopride binding potential due to some intervention can be interpreted as reflecting an intervention-induced change in DA reIease (presuming the modeling has been done correctly, which I am uncertain about in the present case). It cannot be interpreted as reflecting a change in DA receptor 'activity' (whatever that might mean!). I assume that when they say that their manipulation "increases DA receptor activity" what they mean is that they found a reduction in [^11^C]raclopride binding potential (i.e. increased DA release). But how they talk about their findings is conceptually wrong and needlessly confusing to the reader.

In light of this feedback, the reviewers' concern regarding the interpretation and presentation of the PET signals has been further increased. Based on these comments, we concluded that further verification of the technique is required to properly interpret the PET signals, and this might significantly change the authors' interpretations and conclusions.

The reviewers' individual comments are appended below.

Reviewer #1:

Social songbirds, like zebra finches, have to balance tolerance to the surrounding neighbors and selectivity to keep monogamous pairs (social behavior vs sexual behavior). Both for social and sexual behavior, dopaminergic signals are recognized as representing rewards, however, how this neuromodulator acts on the brain circuits is still largely unknown. In this paper, experiments were well designed to answer how dopaminergic activity is regulated differently between sexes and contexts. The authors show there are sexual differences in the dopaminergic activity in the striatum in responses to the song stimulation.

I found this paper exciting and important, as it showed for the first time (as far as I know) that sexual dimorphic response in dopamine activity to the song presentation, depending on birds' mating history. It also provided the new method which can measure the dopamine activity in freely behaving birds, although detailed receptor mechanism, thresholding mechanism etc. have not yet demonstrated. That would explain the brain mechanism to code reward signal depending on the contexts and history of animals' behavior, as well as the possible sexual dimorphism in coding.

The paper is well written and can be read smoothly so that I have some minor concerns stated in the following section.

Reviewer #2:

This manuscript uses PET imaging and pharmacology to test the roles of dopamine-striatal circuits in mediating reinforcing responses to song in the zebra finch. They find that males and females exhibit different dopamine responses and different behavioral responses to male song. Males were willing to sustain mild airpuffs to hear a variety of songs. Females, on the other hand, only appeared to find the song of their mated male rewarding. Striatal dopamine responses in males and females were consistent with these behavioral responses. The authors suggest, but do not demonstrate, that these results could support social behavior among males consistent with a gregarious, but not territorial, society.

Strengths:

1) The assay to test song reinforcement behaviorally is a clever adaptation of addiction paradigms to measure reinforcement in birds. The PET imaging is highly complementary.

2) Figure 1 and Figure 2 provide very clear representations of experimental design, making this a very easy paper to read and understand.

Weaknesses:

1) The main weakness in this study is that dopamine activity is that the authors do not consider or control for the plausible possibility that animal movement is a major contributor to dopamine activity, and a potential confound of their behavioral assays. Specifically, dopamine activity is strongly modulated by movement in mammals (e.g. Jin and Costa, 2010; Howe and Dombeck, 2016;) and in songbirds (Gadagkar et al., 2016). Thus, any stimuli that elicits increased animal movement will also increase striatal dopamine, even if the stimulus itself does not actually act on the dopamine system. This possibility would be relatively easy to control for. One route could be to compare PET dopamine levels in two groups of animals: one which recently underwent a high period of activity and one that did not. If movement does not influence PET DA, then their results will hold. But I find this outcome to be unlikely. Another possibility would be to measure animal movement in their behavioral assays and either modify those methods to ensure that movement across distinct groups is relatively equal, or to regress against a movement parameter to demonstrate that a song stimulus, and not movement, is primarily responsible for the PET signal.

Reviewer #3:

Tokarev and colleagues examined the role of dopamine in social interactions in zebra finch. The authors use "delayed-PET" to measure dopamine release in vivo while the birds were exposed to song playbacks. The authors injected [^11^C]raclopride radiotracer that binds to D2 dopamine receptors before experiment. The birds were then exposed to song playbacks. The authors found that song playback increased the PET signal in the striatum only in males but not in females. Further experiments showed that the PET signal in the striatum increases in female birds only when mated females were exposed to the song of their mated males. The authors also examined the reinforcing properties of songs behaviorally. Unmated males were willing to obtain song stimulation in exchange of mildly aversive air puffs whereas unmated females weren't. The authors found that mated females were willing to obtain the song of their mated males but not that of other males (unfamiliar song). Finally, the authors demonstrate that D2 dopamine receptor blockade impaired the reinforcing property of songs in males.

These results are potentially interesting as it demonstrates positively reinforcing properties of song and the involvement of striatal dopamine in this process. Additionally, in female birds, this effect was observed specifically when mated female listen to their mated males. Nonetheless, I do not fully understand the nature of the PET measurement (thus this requires more explanations).

1) The authors state that they used the delayed-PET to measure "dopaminergic activity" or "D2 receptor activity" but I do not fully understand how this technique works. The authors describe: "When dopamine is released, decrease in radioactive [^11^C]raclopride signal is mediated through direct competition between these two molecules for D2 receptors (Fisher et al., 1995) and as a result of D2 receptors switching from low to high affinity for dopamine but not raclopride (Seeman et al., 1994; Fisher et al., 1995)". My understanding is that there is a basal level of [^11^C]raclopride binding before experimental manipulations (song). This basal binding compete with dopamine released during experimental manipulations. This would mean that dopamine release should decrease [^11^C]raclopride binding to D2 receptors, this would in turn reduce radioactive signals. However, the authors discuss increased radioactive signals as increased dopaminergic activity or D2 receptor activity.

First, it seems confusing to use "dopaminergic activity" or "D2 receptor activity". Please use words that more directly relate to what were actually measured (e.g. radioactive signal).

Second, the interpretation of the PET signal appears to depend on many assumptions regarding what really happens at the receptors and extracelluar space. It is possible that this is well-established in the field, but it is important to more explicitly explain it explicitly. What does the increase in the PET signal really indicate? Does it indicate increased dopamine release or decrease?

[Editors’ note: what now follows is the decision letter after the authors submitted for further consideration.]

Thank you for submitting your article "Sexual dimorphism in striatal dopaminergic responses promotes monogamy in social songbirds" for consideration by *eLife*. Your revised article and letter of appeal have been reviewed by three peer reviewers, and the evaluation has been overseen by a Reviewing Editor and a Senior Editor. The reviewers have opted to remain anonymous.

Summary:

We have considered your appeal, and the revised manuscript was sent to the original reviewers and a PET expert (Please note that the previous Reviewer 3 was replaced by the PET expert). The individual comments are appended below. Overall, the reviewers found that, as in the previous review, the manuscript contains potentially interesting findings. The reviewers thought that the manuscript is improved. The authors now present their analysis on body movements that addresses the previous concern regarding the possibility that dopamine signals may be due to movements. However, the reviewers still raised concerns as to the author's analysis methods and their descriptions of signals. These issues need to be addressed before publication.

Essential revisions:

1) There are still many places where they refer to "dopaminergic activity", "dopamine receptor activity", "D2 receptor activity", "neuronal activity related to dopamine release" and "activation of dopamine receptors". For instance, the authors state that "First, we used a delayed positron emission tomography (PET) procedure (Patel et al., 2008) in order to measure the accumulation of dopaminergic activity (neuronal activity related to dopamine release and activation of dopamine receptors)." The reviewers found that these statements are very confusing. Although we appreciate that the authors reduced some of similar statements from the previous manuscript, we would like the authors to address thoroughly to avoid any confusion. Please refer to Reviewer 3's comment #1 for more detail.

2) The authors compare "silent" versus "song" conditions. Because the author does not compare different time points in the same animal, the authors cannot conclude whether dopamine binding was increased or decreased. What if "silent" condition caused a decrease in [^11^C]raclopride signals? Please avoid using "increase" or "decrease" unless the authors can justify it.

3) The authors stated that they injected D2 receptor antagonist L-741,626 before injecting [^11^C]raclopride "to test whether the song reinforcement we observed in males was driven by striatal D2 receptor activity". However, [^11^C]raclopride radioactivity alone does not support this claim. Please revise it. Please refer to Reviewer #1's comment #2.

4) Reviewer 3 raised a concern regarding the distinction between "group" and "individual" analyses. Please address this point (his/her point #2).

5) Reviewer 3 raised a concern regarding the conversion from striatal-cerebellar ratios to the SOR (striatal occipital ratio) values. Please address this point (his/her point #3).

6) Please provide more information regarding the timing of stimulus presentation relative to [^11^C]raclopride injection time. (Reviewer 3's point #4).

Reviewer #1:

The revised manuscript looks better than the original one, especially in the points of describing detailed methods and additional data on D2 receptor antagonist injection.

However, their data presentation is still confusing and did not show the data directly. Especially, Reviewer 3 asked to use the better term to directly reflect their data, they still use 'D2 receptor activity' without justifying their use of this term. (Even though they stated that they removed the problematic term 'dopamine receptor activity' in their rebuttal letter). In more specific:

Their methods explained that they measure the [^11^C]raclopride radioactivity with PET. That means if DA releases happen [^11^C]raclopride radioactivity decreases by receptor binding competition. They use [^11^C]raclopride radioactivity in the cerebellum as a baseline as there is no D2 receptor expression in there.

In their delayed PET methods, if my understanding is correct, birds were injected with [^11^C]raclopride and exposed to song or silence for 20 min, then scanned [^11^C]raclopride radioactivity by PET. So, they measured the [^11^C]raclopride radioactivity only after song (silent) exposure and it is not possible to measure the [^11^C]raclopride radioactivity changes before and after song listening. However, they use the term 'increase' or 'decrease' of 'D2 receptor activity' and 'change in the dopamine level' which are really confusing about what they measured.

In the study which they provided as a reference for delayed-PET (Marzluff et al., 2012), they measured the radioactivity of [F-18]fluorodeoxyglucose over the time course of the presentation of different visual stimulus and compared them. I think they need more clear justification of their delayed-PET methods.

They stated that they injected D2 receptor antagonist L-741,626 before injecting [^11^C]raclopride 'to test whether the song reinforcement we observed in males was driven by striatal D2 receptor activity' (l139). However, it can test only whether [^11^C]raclopride binding is on D2 receptor, and cannot test whether song reinforcement was driven by striatal D2 receptor (for that they should inject D2 receptor antagonist into the striatum and see the effect on song reinforcement behavior, the air puff experiment). Also, we can see DA binding in the striatum with smaller [^11^C]raclopride radioactivity comparing to the cerebellum. But if D2 receptor antagonist is there, we cannot see the [^11^C]raclopride radioactivity in any condition. What we can expect is only the [^11^C]raclopride radioactivity difference between with saline injection, which tells only that the delayed PET measure is D2 receptor specific (even for that it would be better to test D1 receptor antagonist also as D1A receptor expression is already reported). It should be clear that what this experiment is for.

Reviewer #2:

My main concern with the initial manuscript was that the potential influence of movement on striatal dopamine signals was not addressed. This revision uses 16 new birds to carefully assess movement patterns in response to song. They find that movement is highly unlikely to explain the differences in DA responses between males and females. I still think this is an interesting paper that provides a stong link between striatal dopamine and song-related social behavior.

The entire study depends on the validity of PET for measuring striatal DA. Regrettably I lack the specific expertise to weigh in on whether or not this revision adequately addresses the legitimate concerns of the PET expert. If it is determined that their measurements are valid, then I could support this paper. But if it is determined that the interpretations of the PET data are an overreach, then I would defer to the PET expert and support her/his decision.

Reviewer #3:

It is definitely an improvement. My original comments are not much changed. On balance, my sense is that the data are ok, but there are still a few things that bother me regarding the PET component of the study.

1) How they interpret and discuss condition-induced changes in [^11^C]raclopride binding. There are still many places where they refer to "dopaminergic activity" or dopamine receptor activity. Dopamine "transmission" is a more accurate way of phrasing this. In a few places, the way that their language just doesn't make sense to me. For example: "First, we used a delayed positron emission tomography (PET) procedure (Patel et al., 2008) in order to measure the accumulation of dopaminergic activity (neuronal activity related to dopamine release and activation of dopamine receptors)." PET doesn't measure the accumulation of "dopaminergic activity" and it definitely doesn't measure neuronal activity or the activation of dopamine receptors (though some of the PET signal may be due to agonist-induced internalization of DA receptors, I don't think that's what they're referring to or mean here). As another example: "confirming that D2 receptor activity is a robust indicator of the overall striatal dopamine release." They're not measuring D2 receptor activity; they're measuring condition-induced changes in D2 receptor availability. The discussion still has many references to receptor activity.

2) They quantify condition-induced changes in [^11^C] raclopride binding by extracting binding estimates from a region of interest for each subject from the group-averaged parameter map. This is not a problem. They make a distinction between "group" and "individual" analyses that is incorrect, because they subsequently submit those individual values to a group-level statistical contrast. This is fine as a means of quantifying the average change in [^11^C]raclopride binding. However, it is no more an "individual" analysis than the original group-level (imaging) contrast from which the individual values were derived. Here's an example of this:

“We detected a cluster of voxels with significantly lower [^11^C]raclopride binding in response to mate song in a small part of the medial dorsal striatum (Figure 8). At the group level, the difference across those voxels did not survive correction for multiple comparisons (Figure 8). Nevertheless, at the individual level, the same area did show a statistically significant 12{plus minus}4% decrease in [^11^C]raclopride binding to mate song compared to non-mate song (Figure 8=0.042, paired t-test).”

This example is especially problematic because they're using the "individual level" analysis for inference. They didn't get a significant result from the whole-brain contrast, which is – appropriately – corrected for multiple comparisons, so they extracted signal from the non-significant cluster identified from that contrast and re-ran the analysis outside of imaging space. It is just under p<0.05, so they report it as significant. This practice is considered invalid in imaging because of what has been called circularity, non-independence, or double-dipping (see Vul 2008, Kriegskorte, 2009).

3) I am confused by their conversion from striatal-cerebellar ratios to the SOR (striatal occipital ratio) values displayed in the figures and apparently used for inference. They cite Patel, 2008, as justification for this, but Patel, 2008, never mention SOR values. I'm honestly not sure why they did this, but it is odd.

4) I would have liked more information regarding the timing of stimulus presentation relative to [^11^C]raclopride injection time. They mention [^11^C]raclopride half-life as their reason for selecting 20 minutes, but stimulus timing onset should be calibrated to an estimate of striatal D2 receptor saturation (i.e. the stimulus should be introduced once equilibrium is reached with [^11^C] raclopride). Again, this is one of those things that may very well be ok, but I was looking for more information/justification about the timing choice.

---

## [Author Response]

[Editors’ note: the author responses to the first round of peer review follow.]

We thank the three reviewers and the expert of PET techniques for their suggestions and criticism, all of which were addressed in this revised version provided with the rebuttal, which, we feel, has improved greatly in the process. In particular, we addressed the concern about movement artifacts by presenting new movement tracking data, and the PET technique is now explained in much more details, using standard terminology.

*[…] Typically, a parametric pharmacokinetic model is used to derive a measure of non-displaceable [^11^C]raclopride binding potential for each voxel or region of interest. This is used as a proxy for "D2/D3 receptor availability." Because DA competes with [^11^C]raclopride for available D2/D3 receptors, a reduction in [^11^C]raclopride binding potential following a cognitive/behavioral or pharmacological challenge is taken as evidence for DA release. For example, administering amphetamine will cause striatal [^11^C]raclopride binding potential to decrease (compared to administration of placebo). In the few studies that have also obtained simultaneous direct measures of DA release (typically through microdialysis), the magnitude of stimulus-induced striatal DA release is strongly correlated with the size of the reduction in [^11^C]raclopride binding potential after stimulant administration.*

*With all that said, the pharmacokinetic models are not particularly robust. Practically, this places significant constraints on experimental design. I raise this point because it seems that the authors used a "delayed PET" approach that is highly nonstandard.*

Indeed, delayed PET techniques are rarely used in human studies. However, this methodology is well established in freely moving non-human animals. This method is very useful as it prevents immobilization stress, which is known to interfere with the striatal dopaminergic system. Delayed PET technique was successfully implemented in rodents about a decade ago (Patel et al., 2008). Since then, it has been replicated in multiple species, including songbirds (Marzluff et al., 2012). Usage of [^11^C]raclopride to track changes in dopamine levels has been validated in studies with simultaneous microdialysis (Morris et al., 2008; Normandin et al., 2012). As we mentioned in the text, the changes observed in our work were in the same range as changes in dopamine levels measured by microdialysis in another study on zebra finches (Ihle et al., 2015). Finally, the idea of using PET to assess changes in striatal dopamine during rewarding auditory stimuli is not new either: a Nature Neuroscience paper from 2011 showed, in humans, that music-triggered changes in striatal dopamine (as measured by [^11^C]raclopride PET), strongly correlated with the amount of pleasure subjects reported (Salimpoor et al., 2011). Our study is very similar, except for the 20-minute delay in the scan (as per ‘delayed PET’ protocol). In the revised manuscript we have added these and other references in order to clarify that delayed PET scan methodology is well established in nonhuman animals.

*In human PET imaging, investigators will typically inject [^11^C]raclopride, measure signal until the tracer reaches equilibrium to obtain a pre-challenge binding potential estimate, introduce the cognitive/behavioral/drug challenge, and then measure signal again to get a post-challenge binding potential estimate. It's this binding potential delta (reflecting the degree to which a challenge reduces binding potential) that's then compared between groups or correlated with some individual difference measure. In principle, it's correct to interpret a challenge-induced increase in [^11^C]raclopride binding potential as reflecting a challenge-induced decrease in DA. In practice, though, this is uncommon and such a finding would either have to be supported by corroborative measures or a compelling, a-priori mechanistic prediction. I should say that their design may be entirely appropriate. What concerning is that they don't provide a coherent justification for this non-standard design, nor do they offer any validation that their models fit the data well given this design.*

We did offer validations. In fact, our work goes beyond other studies providing validations by combining three different methods: PET, behavioural reinforcement, and pharmacological D2-receptor blockage. All three methods point to the same phenomenon, e.g., the PET results suggest that dopamine level might increase during song playback, but only in males. The behavioural results show behavioural reinforcement only in males. Finally, blocking D2 receptors blocked the effect in males. It took us several years to establish those validations, and we feel that requiring us, in addition, to re-establish the validity on the delayed PET technique at the basic level should not be necessary, given previously published and our current validations.

*The way the authors talk about PET measurements is incredibly confusing. [^11^C]raclopride PET does not measure "dopamine receptor activity;" it measures [^11^C]raclopride binding potential! A change in [^11^C]raclopride binding potential due to some intervention can be interpreted as reflecting an intervention-induced change in DA reIease (presuming the modeling has been done correctly, which I am uncertain about in the present case). It cannot be interpreted as reflecting a change in DA receptor 'activity' (whatever that might mean!). I assume that when they say that their manipulation "increases DA receptor activity" what they mean is that they found a reduction in [^11^C]raclopride binding potential (i.e. increased DA release). But how they talk about their findings is conceptually wrong and needlessly confusing to the reader.*

*In light of this feedback, the reviewers' concern regarding the interpretation and presentation of the PET signals has been further increased. Based on these comments, we concluded that further verification of the technique is required to properly interpret the PET signals, and this might significantly change the authors' interpretations and conclusions.*

The reviewers' individual comments are appended below.

We sincerely apologize for the confusion caused by the terminology that we introduced in the paper. We have now reverted to using the standard terminology. We acknowledge that our documentation of the PET methodology was deficient. We have therefore revised the manuscript to thoroughly describe the method. We have also removed the problematic term ‘dopamine receptor activity’ and replaced it with the more appropriate term ‘[^11^C]raclopride binding’. We would like to explain here why we introduced a new terminology, and how it, unfortunately, ended up confusing the expert and Reviewer #3. We were worried that the technical term ‘[^11^C]raclopride binding’ could confuse readers that are unfamiliar with PET methodology, since one has to keep reversing the logic: less [^11^C]raclopride binding suggests an increase in dopamine level. Hence, we modified the formula to capture the negative relation between dopamine levels and the raclopride signal. But we ended up confusing readers who were familiar with PET methodology. We suspect that the PET expert did not notice the change we made to the formula (which we only presented in the Materials and methods section), as the claim "In principle, it's correct to interpret a challenge-induced increase in [^11^C]raclopride binding potential as reflecting a challenge-induced decrease in DA. In practice, though, this is uncommon…” does not apply to our study. As in all other papers on dopamine PET, we interpreted decrease in [^11^C]raclopride binding as reflecting an increase in DA, but we presented the data with the reversed formula. Given the inadvertent confusion this caused for readers familiar with PET, we are of course switching to the traditional terminology in the revised manuscript.

*Reviewer #1:*

*Social songbirds, like zebra finches, have to balance tolerance to the surrounding neighbors and selectivity to keep monogamous pairs (social behavior vs sexual behavior). Both for social and sexual behavior, dopaminergic signals are recognized as representing rewards, however, how this neuromodulator acts on the brain circuits is still largely unknown. In this paper, experiments were well designed to answer how dopaminergic activity is regulated differently between sexes and contexts. The authors show there are sexual differences in the dopaminergic activity in the striatum in responses to the song stimulation. I found this paper exciting and important, as it showed for the first time (as far as I know) that sexual dimorphic response in dopamine activity to the song presentation, depending on birds' mating history. It also provided the new method which can measure the dopamine activity in freely behaving birds, although detailed receptor mechanism, thresholding mechanism etc. have not yet demonstrated. That would explain the brain mechanism to code reward signal depending on the contexts and history of animals' behavior, as well as the possible sexual dimorphism in coding.*

We thank the reviewer for those kind words. We added a paragraph to the discussion, to highlight the limitations of our techniques and to present open questions about cellular/molecular mechanisms. There are several potential mechanisms that could account for the sexual dimorphism we observed, including different receptors expression levels, different densities of dopaminergic cells, different ratios of D1/D2 receptors and different reuptake mechanisms. Those are now mentioned in the revised discussion.

*The paper is well written and can be read smoothly so that I have some minor concerns stated in the following section.*

*Reviewer #2:*

*This manuscript uses PET imaging and pharmacology to test the roles of dopamine-striatal circuits in mediating reinforcing responses to song in the zebra finch. They find that males and females exhibit different dopamine responses and different behavioral responses to male song. Males were willing to sustain mild airpuffs to hear a variety of songs. Females, on the other hand, only appeared to find the song of their mated male rewarding. Striatal dopamine responses in males and females were consistent with these behavioral responses. The authors suggest, but do not demonstrate, that these results could support social behavior among males consistent with a gregarious, but not territorial, society.*

We thank the reviewer for clarifying this point: the sexual dimorphism that we found suggests a mechanism that can potentially explain the coexistence of gregariousness and monogamy, but we should be careful not to overstate this idea, as it should be tested further in future studies. In the revised version we added a discussion paragraph about how cross species PET studies could further test this hypothesis.

Strengths:

*1) The assay to test song reinforcement behaviorally is a clever adaptation of addiction paradigms to measure reinforcement in birds. The PET imaging is highly complementary.*

*2) Figure 1 and Figure 2 provide very clear representations of experimental design, making this a very easy paper to read and understand.*

Weaknesses:

*1) The main weakness in this study is that dopamine activity is that the authors do not consider or control for the plausible possibility that animal movement is a major contributor to dopamine activity, and a potential confound of their behavioral assays.*

We were aware of this issue, and we did mention that in our experimental conditions, our birds did not move much. But we agree with the reviewer that even slight movements could potentially affect striatal dopamine. We added a new control group where we continuously tracked body and head movement to directly test if movement artifacts can explain our results (Figure 3—figure supplement 2; Table 1 and Table 2).

*Specifically, dopamine activity is strongly modulated by movement in mammals (e.g. Jin and Costa, 2010; Howe and Dombeck, 2016;) and in songbirds (Gadagkar et al., 2016). Thus, any stimuli that elicits increased animal movement will also increase striatal dopamine, even if the stimulus itself does not actually act on the dopamine system. This possibility would be relatively easy to control for.*

We agree that if movement was elicited by presenting auditory stimuli we would need to single it out to see what actually caused the effect on the dopamine system.

*One route could be to compare PET dopamine levels in two groups of animals: one which recently underwent a high period of activity and one that did not. If movement does not influence PET DA, then their results will hold. But I find this outcome to be unlikely. Another possibility would be to measure animal movement in their behavioral assays and either modify those methods to ensure that movement across distinct groups is relatively equal, or to regress against a movement parameter to demonstrate that a song stimulus, and not movement, is primarily responsible for the PET signal.*

We are grateful for these suggestions, and we followed the second one in the revised manuscript. If dopamine level change were due to movement, then movement should differ across treatments and sexes: higher in male zebra finches when hearing songs compared to when they are kept in silence, but not so in females. To test if this were the case, we performed an additional control experiment with a new group of 8 males and 8 females, where we simulated the song vs. silence PET conditions (including transfer and non-radioactive ligand injection), and also video tracked birds’ movement. Even with a continuous account of all body movements (e.g., flying, hopping and wing-whirring) throughout the session, we failed to find any significant changes in the amount of movement across treatments. Although head movement was higher during song playbacks, there was no significant difference between males and females. Therefore, movement artifacts cannot explain the effect of songs on dopamine that we observed. We include these results in the revised manuscript (Figure 3—figure supplement 2; Table 1 and Table 2).

*Reviewer #3:*

*Tokarev and colleagues examined the role of dopamine in social interactions in zebra finch. The authors use "delayed-PET" to measure dopamine release in vivo while the birds were exposed to song playbacks. The authors injected [^11^C]raclopride radiotracer that binds to D2 dopamine receptors before experiment. The birds were then exposed to song playbacks. The authors found that song playback increased the PET signal in the striatum only in males but not in females. Further experiments showed that the PET signal in the striatum increases in female birds only when mated females were exposed to the song of their mated males. The authors also examined the reinforcing properties of songs behaviorally. Unmated males were willing to obtain song stimulation in exchange of mildly aversive air puffs whereas unmated females weren't. The authors found that mated females were willing to obtain the song of their mated males but not that of other males (unfamiliar song). Finally, the authors demonstrate that D2 dopamine receptor blockade impaired the reinforcing property of songs in males.*

*These results are potentially interesting as it demonstrates positively reinforcing properties of song and the involvement of striatal dopamine in this process. Additionally, in female birds, this effect was observed specifically when mated female listen to their mated males. Nonetheless, I do not fully understand the nature of the PET measurement (thus this requires more explanations).*

*1) The authors state that they used the delayed-PET to measure "dopaminergic activity" or "D2 receptor activity" but I do not fully understand how this technique works. The authors describe: "When dopamine is released, decrease in radioactive [^11^C]raclopride signal is mediated through direct competition between these two molecules for D2 receptors (Fisher et al., 1995) and as a result of D2 receptors switching from low to high affinity for dopamine but not raclopride (Seeman et al., 1994; Fisher et al., 1995)". My understanding is that there is a basal level of [^11^C]raclopride binding before experimental manipulations (song). This basal binding compete with dopamine released during experimental manipulations. This would mean that dopamine release should decrease [^11^C]raclopride binding to D2 receptors, this would in turn reduce radioactive signals.*

Yes, this is correct. However, we were worried that the technical term ‘[^11^C]raclopride binding’ could confuse most readers (who are unfamiliar with PET methodology), since one has to keep reversing the logic: less [^11^C]raclopride binding suggests an increase in dopamine level. Hence, we modified the formula to capture the negative relation between dopamine levels and the raclopride signal. This simple modification in the formula was designed to show changes in dopamine levels directly rather than inversely. This modification was described in the last paragraph of the Materials and methods section, but neither this reviewer nor the PET expert noticed it. This unfortunate decision of ours is the source of the confusion, which we have fixed in the revised version of the MS, where the PET technique is describe with more details and without deviating from the standard terminology used in the PET literature.

*However, the authors discuss increased radioactive signals as increased dopaminergic activity or D2 receptor activity.*

Not at all! Like everyone else, we interpret decrease in the PET signal as an increase in dopamine. So, as the radioactive signal is inversely related to dopaminergic activity, we changed the formula in order to capture this notion, and came up with the term ‘dopaminergic activity’, to make it easer to interpret the figures. Unfortunately, this led to confusion, and therefore, we have now decided to abandon this idea and in the revised manuscript present the PET data without modifying the formula.

*First, it seems confusing to use "dopaminergic activity" or "D2 receptor activity". Please use words that more directly relate to what were actually measured (e.g. radioactive signal).*

We agree and have followed this advice in the revised manuscript. We now use the technical term ‘[^11^C]raclopride binding’ and have abandoned the change in the formula that we did in the original manuscript. We apologize for this inadvertent confusion and we are, of course, switching to the traditional terminology in the revised manuscript. We now define the term “dopaminergic activity” in the introduction, in order to facilitate the discussion of dopaminergic mechanisms in the abstract, introduction and discussion, as following: neuronal activity related to dopamine release and activation of dopamine receptors.

*Second, the interpretation of the PET signal appears to depend on many assumptions regarding what really happens at the receptors and extracelluar space. It is possible that this is well-established in the field, but it is important to more explicitly explain it explicitly. What does the increase in the PET signal really indicate? Does it indicate increased dopamine release or decrease?*

Yes, decrease in PET signal reflects increased dopamine release. Note our statement in the original manuscript (subsection “PET image preparation and statistical analysis”): “As [^11^C]raclopride and dopamine compete for D2-receptors, decrease in radioactive signal indicates an increase of dopamine concentration (Fisher et al., 1995; Endres et al., 1997).” As we explain in the replies to the previous comments, we now use standard PET terminology and formula. We followed the reviewer suggestion, and in the revised manuscript we have also included an additional description of our methodology.

[Editors’ note: the author responses to the re-review follow.]

*Essential revisions:*

*1) There are still many places where they refer to "dopaminergic activity", "dopamine receptor activity", "D2 receptor activity", "neuronal activity related to dopamine release" and "activation of dopamine receptors". For instance, the authors state that "First, we used a delayed positron emission tomography (PET) procedure (Patel et al., 2008) in order to measure the accumulation of dopaminergic activity (neuronal activity related to dopamine release and activation of dopamine receptors)." The reviewers found that these statements are very confusing. Although we appreciate that the authors reduced some of similar statements from the previous manuscript, we would like the authors to address thoroughly to avoid any confusion. Please refer to Reviewer 3's comment #1 for more detail.*

We have removed the term ‘dopaminergic activity’ and, following advice of Reviewer 3, replaced it by the term “dopamine neurotransmission”. Also, when referring to figures and data directly, we now consistently use the term [^11^C]raclopride binding. The revised manuscript uses the same terminology as in Salimpoor e al., 2011.

*2) The authors compare "silent" versus "song" conditions. Because the author does not compare different time points in the same animal, the authors cannot conclude whether dopamine binding was increased or decreased. What if "silent" condition caused a decrease in [^11^C]raclopride signals? Please avoid using "increase" or "decrease" unless the authors can justify it.*

We omitted these terms and refer to differences as “higher/lower levels of [^11^C]raclopride binding” when referring to figures and data directly, and “higher/lower levels of dopamine neurotransmission” elsewhere. We agree that our silent condition is not biologically ‘neutral’, it is just a reasonable baseline for assessing the effect of song playbacks on dopamine neurotransmission.

*3) The authors stated that they injected D2 receptor antagonist L-741,626 before injecting [^11^C]raclopride "to test whether the song reinforcement we observed in males was driven by striatal D2 receptor activity". However, [^11^C]raclopride radioactivity alone does not support this claim. Please revise it. Please refer to Reviewer #1's comment #2.*

Indeed [^11^C]raclopride signal alone would not have been enough to prove our claim, but the experiment with D2 receptor antagonist L-741,626 consisted of two parts, and in the second part we did provide evidence from the behavioral preference test while under the influence of D2 receptor antagonist. To make it clearer that these are parts of the same experiment, we have now combined those paragraphs into one, and rephrased the aim more accurately:

“To test whether the song reinforcement we observed in males was dependent on dopamine neurotransmission, we used the D2 receptor antagonist L-741,626 to interfere with D2 receptors.”

*4) Reviewer 3 raised a concern regarding the distinction between "group" and "individual" analyses. Please address this point (his/her point #2).*

The reviewer is correct; these were not separate statistical analyses. We have now clarified it by changing the wording. We have replaced the term ‘individual analysis’ with ‘exploratory post-hoc analysis’. In the mated females group, where the cluster identified was not significant after adjustment but post-hoc analysis showed evidence to increase signal in that area (with p=0.04), we changed the wording to indicate that the p-value obtained should not be interpreted as statistically significant, but as evidence for a weak effect that should be validated in future studies.

*5) Reviewer 3 raised a concern regarding the conversion from striatal-cerebellar ratios to the SOR (striatal occipital ratio) values. Please address this point (his/her point #3).*

We are sorry for this error. The cerebellum was used as a reference area throughout our study. The correct term is “striatal-cerebellar ratio”.

*6) Please provide more information regarding the timing of stimulus presentation relative to [^11^C]raclopride injection time. (Reviewer 3's point #4).*

We provided the information on the onset of stimulation within 1–2 minutes of recovery after injection (subsection “Simultaneous PET on four zebra finches to measure dopamine released during auditory stimulation in awake unrestrained state”). We now have backed up this timing by citing two previous studies that used “delayed PET” method (subsection “Simultaneous PET on four zebra finches to measure dopamine released during auditory stimulation in awake unrestrained state”).).

*Reviewer #1:*

*The revised manuscript looks better than the original ones, especially in the points of describing detailed methods and additional data on D2 receptor antagonist injection.*

*However, their data presentation is still confusing and did not show the data directly. Especially, Reviewer 3 asked to use the better term to directly reflect their data, they still use 'D2 receptor activity' without justifying their use of this term. (Even though they stated that they removed the problematic term 'dopamine receptor activity' in their rebuttal letter). In more specific:*

This is now fixed. Just to clarify, in the previous submission we did remove the term “dopaminergic activity” from the Results, but defined it the Introduction, and used it only in the Discussion. We now have removed it entirely and following advice of Reviewer 3 have replaced it with “dopamine neurotransmission”. When referring to data directly we now use the term “[^11^C]raclopride binding”. This terminology is adapted from Salimpoor et al., (2011).

*Their methods explained that they measure the [^11^C]raclopride radioactivity with PET. That means if DA releases happen [^11^C]raclopride radioactivity decreases by receptor binding competition. They use [^11^C]raclopride radioactivity in the cerebellum as a baseline as there is no D2 receptor expression in there.*

Yes, we now refer to this measurement as the “striatal–cerebellar ratio”.

*In their delayed PET methods, if my understanding is correct, birds were injected with [^11^C]raclopride and exposed to song or silence for 20 minutes, then scanned [^11^C]raclopride radioactivity by PET. So, they measured the [^11^C]raclopride radioactivity only after song (silent) exposure and it is not possible to measure the [^11^C]raclopride radioactivity changes before and after song listening. However, they use the term 'increase' or 'decrease' of 'D2 receptor activity' and 'change in the dopamine level' which are really confusing about what they measured.*

This is a correct description, and we thank the reviewer for pointing out this poor choice of words when describing our results. We now refer to these differences as “higher/lower levels of [^11^C]raclopride binding” (or dopamine neurotransmission).

*In the study which they provided as a reference for delayed-PET (Marzluff et al., 2012), they measured the radioactivity of [F-18]fluorodeoxyglucose over the time course of the presentation of different visual stimulus and compared them. I think they need more clear justification of their delayed-PET methods.*

In Marzluff et al., the crows were presented with only one type of stimulus: either “caring” masks or “threatening” masks in 1-minuteon/off blocks for 14 min. We also presented stimulus of one type before the scan (one song every 15 sec during 20 min). In contrast to our work, each crow in Marzluff et al., study was assigned a treatment at random and scanned only once. We went further and did two scans on the same animals with different types of stimulation, which allowed us to do within–subject analyses. Other than that and a different ligand, we used a very similar protocol.

*They stated that they injected D2 receptor antagonist L-741,626 before injecting [^11^C]raclopride 'to test whether the song reinforcement we observed in males was driven by striatal D2 receptor activity' (Results section). However, it can test only whether [^11^C]raclopride binding is on D2 receptor, and cannot test whether song reinforcement was driven by striatal D2 receptor (for that they should inject D2 receptor antagonist into the striatum and see the effect on song reinforcement behavior, the air puff experiment).*

There is some confusion here: The [^11^C]raclopride signal alone would not have been enough to support our claim, but the experiment with D2 receptor antagonist L-741,626 actually consisted of two parts, and the second part provided evidence from the behavioral preference test. To clarify that these are two parts of the same experiment, we have now combined those paragraphs into one. We also now use a phrase that described the aim more accurately:

“To test whether the song reinforcement we observed in males was dependent on the striatal dopamine neurotransmission, we used the D2 receptor antagonist L-741,626 to interfere with D2 receptor activity.”

So, the first part of the experiment provided us with evidence that this blocker was indeed specific to D2 receptors (same type that we observed in our PET scans), and we saw the effect in the striatum. The second part confirmed that without activity of these receptors there was no rewarding effect by hearing songs in males.

*Also, we can see DA binding in the striatum with smaller [^11^C]raclopride radioactivity comparing to the cerebellum. But if D2 receptor antagonist is there, we cannot see the [^11^C]raclopride radioactivity in any condition. What we can expect is only the [^11^C]raclopride radioactivity difference between with saline injection, which tells only that the delayed PET measure is D2 receptor specific (even for that it would be better to test D1 receptor antagonist also as D1A receptor expression is already reported). It should be clear that what this experiment is for.*

We now clarify that this first part of the experiment is aimed only to test if the antagonist can block the PET signal in the striatum (which it does). The second experiment shows the behavioral effect of this blockage (diminishing the reinforcement effect of song playbacks).

*Reviewer #2:*

*My main concern with the initial manuscript was that the potential influence of movement on striatal dopamine signals was not addressed. This revision uses 16 new birds to carefully assess movement patterns in response to song. They find that movement is highly unlikely to explain the differences in DA responses between males and females. I still think this is an interesting paper that provides a stong link between striatal dopamine and song-related social behavior.*

We thank the reviewer for these kind words, and for the previous suggestions to corroborate our findings by testing the effect of movement directly.

*The entire study depends on the validity of PET for measuring striatal DA. Regrettably I lack the specific expertise to weigh in on whether or not this revision adequately addresses the legitimate concerns of the PET expert. If it is determined that their measurements are valid, then I could support this paper. But if it is determined that the interpretations of the PET data are an overreach, then I would defer to the PET expert and support her/his decision.*

We believe that we have now addressed all the remaining concerns regarding our PET methodology and data interpretation.

*Reviewer #3:*

*It is definitely an improvement. My original comments are not much changed. On balance, my sense is that the data are ok, but there are still a few things that bother me regarding the PET component of the study.*

We thank the reviewer for the positive assessment. We have thoroughly addressed all remaining concerns regarding our PET methodology and data interpretation as elaborated below.

*1) How they interpret and discuss condition-induced changes in [^11^C]raclopride binding. There are still many places where they refer to "dopaminergic activity" or dopamine receptor activity. Dopamine "transmission" is a more accurate way of phrasing this. In a few places, the way that their language just doesn't make sense to me. For example: "First, we used a delayed positron emission tomography (PET) procedure (Patel et al., 2008) in order to measure the accumulation of dopaminergic activity (neuronal activity related to dopamine release and activation of dopamine receptors)." PET doesn't measure the accumulation of "dopaminergic activity" and it definitely doesn't measure neuronal activity or the activation of dopamine receptors (though some of the PET signal may be due to agonist-induced internalization of DA receptors, I don't think that's what they're referring to or mean here). As another example: "confirming that D2 receptor activity is a robust indicator of the overall striatal dopamine release." They're not measuring D2 receptor activity; they're measuring condition-induced changes in D2 receptor availability. The discussion still has many references to receptor activity.*

Following the reviewer suggestion we replaced ‘dopaminergic activity with “dopamine neurotransmission”.

When referring directly to figures or data, we now use the term “[^11^C]raclopride binding” consistently.

*2) They quantify condition-induced changes in [^11^C] raclopride binding by extracting binding estimates from a region of interest for each subject from the group-averaged parameter map. This is not a problem. They make a distinction between "group" and "individual" analyses that is incorrect, because they subsequently submit those individual values to a group-level statistical contrast. This is fine as a means of quantifying the average change in [^11^C]raclopride binding. However, it is no more an "individual" analysis than the original group-level (imaging) contrast from which the individual values were derived. Here's an example of this:*

*“We detected a cluster of voxels with significantly lower [^11^C]raclopride binding in response to mate song in a small part of the medial dorsal striatum (Figure 8). At the group level, the difference across those voxels did not survive correction for multiple comparisons (Figure 8). Nevertheless, at the individual level, the same area did show a statistically significant 12 +/- 4% decrease in [^11^C]raclopride binding to mate song compared to non-mate song (Figure 8=0.042, paired t-test).”*

*This example is especially problematic because they're using the "individual level" analysis for inference. They didn't get a significant result from the whole-brain contrast, which is – appropriately – corrected for multiple comparisons, so they extracted signal from the non-significant cluster identified from that contrast and re-ran the analysis outside of imaging space. It is just under p<0.05, so they report it as significant. This practice is considered invalid in imaging because of what has been called circularity, non-independence, or double-dipping (see Vul 2008, Kriegskorte, 2009).*

Yes, we completely agree, and we have changed the wording so it does not sound like those were separate statistical analyses. We no longer say that we look at the data either at the group or individual level, and simply provide individual data points as exploratory post–hoc analysis. In this particular example, we now say that it is only “a trend for higher levels of dopamine transmission in response to mates’ songs in females” and we clarify that the p value of 0.04 in the should be interpreted as statistically significant, but as suggesting a weak effect in those females that should be further tested in future studies (Results section, Discussion section).

*3) I am confused by their conversion from striatal-cerebellar ratios to the SOR (striatal occipital ratio) values displayed in the figures and apparently used for inference. They cite Patel, 2008, as justification for this, but Patel, 2008, never mention SOR values. I'm honestly not sure why they did this, but it is odd..*

We are sorry for this mistake. We use the cerebellum as a reference area throughout the study. The correct term is, of course, “striatal–cerebellar ratio”.

*4) I would have liked more information regarding the timing of stimulus presentation relative to [^11^C]raclopride injection time. They mention [^11^C]raclopride half-life as their reason for selecting 20 minutes, but stimulus timing onset should be calibrated to an estimate of striatal D2 receptor saturation (i.e. the stimulus should be introduced once equilibrium is reached with [^11^C] raclopride). Again, this is one of those things that may very well be ok, but I was looking for more information/justification about the timing choice.*

Animals were presented with stimulation within 1–2 minutes after injection for 20 minutes (subsection “Simultaneous PET on four zebra finches to measure dopamine released during auditory stimulation in awake unrestrained state”).

We now have backed up this timing by citing two previous studies that used “delayed PET” method (subsection “Simultaneous PET on four zebra finches to measure dopamine released during auditory stimulation in awake unrestrained state”), as our approach is similar to that of Patel el al., 2008 (immediately after injection for 30 minutes) and Marzluff et al., 2012 (immediately after injection for 14 minutes).